# E-RespiNet: An LLM-ELECTRA driven triple-stream CNN with feature fusion for asthma classification

Mohammed Tawfik[1]*, Islam S. Fathi[2,4], Sunil S. Nimbhore[3], Issa M. Alsmadi[2], Mohamed S. Sawah[4]

**1** Faculty of Computer and Information Technology, Sana'a University, Sana'a, Yemen, **2** Department of Computer Science, Faculty of Information Technology, Ajloun National University, Ajloun, Jordan, **3** Department of Computer Science & IT Dr. Babasaheb Ambedkar Marathwada University, Aurangabad (MS), India, **4** Department of Information Systems, Al Alson Higher Institute, Cairo, Egypt

☻ These authors contributed equally to this work.
* kmkhol01@gmail.com

## Abstract

Respiratory disease diagnosis remains challenging in resource-constrained settings, where limited specialist expertise contributes to diagnostic uncertainties affecting over 300 million people worldwide. This study presents E-RespiNet, a novel multi-modal deep learning architecture that integrates ELECTRA's discriminative pre-training with a triple-stream convolutional neural network framework, enhanced through Harmony Search with Opposition-Based Learning optimization for automated respiratory sound classification. The architecture simultaneously processes mel-frequency cepstral coefficients, discrete wavelet transforms, and mel-spectrograms through parallel CNN streams, with features integrated through hierarchical fusion and ELECTRA-based contextual enhancement. Comprehensive evaluations on two independent clinical datasets—the Asthma Detection Dataset Version 2 (1,211 recordings across five conditions) and King Abdullah University Hospital dataset (940 samples from 81 subjects across four conditions)—demonstrated exceptional performance with 98.9% and 98.8% accuracy respectively, representing 5.0% and 4.3% improvements over baseline configurations. Cross-institutional validation revealed 75.7% average accuracy with a 23.3% generalization gap, substantially better than typical medical AI cross-domain performance. Gradient-weighted class activation mapping provided clinically relevant interpretability, while the Harmony Search optimization framework enhanced both performance and cross-institutional robustness. These results demonstrate significant advances in automated respiratory sound analysis through discriminative language model integration and metaheuristic optimization, establishing important benchmarks for deployable respiratory diagnostic tools in diverse healthcare settings.

**Data availability statement:** All relevant data are fully available without restriction. The datasets analyzed in this study are publicly accessible: The Asthma Detection Dataset Version 2 is available from the Kaggle platform at https://doi.org/10.34740/kaggle/dsv/9277143. The King Abdullah University Hospital (KAUH) dataset is available from the Mendeley Data repository at https://data.mendeley.com/datasets/jwyy9np4gv/3.

**Funding:** The author(s) received no specific funding for this work.

**Competing interests:** The authors have declared that no competing interests exist.

# 1 Introduction

Asthma is a chronic respiratory condition that affects over 300 million people globally, and its prevalence is expected to increase to 400 million by 2025 [1]. The disease poses a significant healthcare challenge, with an estimated economic burden of $82 billion [2]. Asthma disproportionately affects low- and middle-income countries, where up to 90% of cases may be uncontrolled [3]. This condition impacts quality of life, productivity, and places strain on healthcare systems. Current treatment strategies include inhaled corticosteroids and bronchodilators; however, no definitive cure has been established. Future research should focus on personalized medicine, biologics, and digital health technologies. Global initiatives, such as the Global Initiative for Asthma (GINA), aim to improve asthma care accessibility and reduce disease burden worldwide.

An early and accurate diagnosis is crucial for effective asthma management. Traditional methods rely on clinical expertise, pulmonary function tests, and manual auscultation; however, there is no gold standard test for asthma. Overdiagnosis and underdiagnosis are common, with overdiagnosis potentially affecting up to 30–40% of cases, leading to unnecessary treatment and delayed diagnosis of other conditions [4]. The increasing prevalence of asthma, particularly in resource-limited regions, necessitates the use of more accessible diagnostic tools. Asthma affects approximately 10% of Europeans, and its diagnosis often relies on clinical judgment because of limited access to objective tests in primary care. The increasing prevalence of asthma, particularly in developing regions where the specialist-to-patient ratio can be as low as 1:100,000, necessitates the development of automated and reliable diagnostic tools.

Recent advances in artificial intelligence, particularly deep learning approaches, such as Convolutional Neural Networks (CNNs), have shown promising results in respiratory sound analysis [5,6]. These methods have progressed from simple binary classification to more sophisticated architectures capable of identifying various respiratory conditions. The emergence of large language models has revolutionized medical AI applications, with models such as Med-PaLM and Clinical-BERT demonstrating remarkable performance in clinical text analysis and diagnostic reasoning. However, the adaptation of LLM capabilities for medical audio analysis remains underexplored, representing a significant research gap. ELECTRA's discriminative pretraining approach offers unique advantages over generative medical LLMs for acoustic pattern recognition tasks. Unlike models optimized for text generation or masked token prediction, ELECTRA's discriminator architecture develops an enhanced sensitivity to subtle pattern variations, making it particularly suitable for pathological acoustic signature detection in respiratory sounds.

CNNs have demonstrated effectiveness in extracting meaningful patterns from respiratory acoustics, achieving accuracies between 85% and 90% in controlled settings [7]. Studies have explored the use of spectrograms, Mel-Frequency Cepstral Coefficients (MFCC), and other audio features as inputs for deep learning models [8]. Although these approaches show promise for enhancing the early detection

and remote monitoring of respiratory diseases, challenges remain, including feature interpretation, generalization across diverse patient populations, and robustness to background noise.

The emergence of Large Language Models (LLMs) has revolutionized various domains of artificial intelligence, yet their potential in medical acoustic analysis remains largely unexplored [9]. Existing methods typically focus solely on acoustic feature extraction and overlook the potential benefits of cross-modal learning and feature enhancement through language-model integration. ELECTRA, with its discriminative pre-training approach and robust feature representation capabilities, offers unique advantages in medical signal processing that remain untapped [10]. This gap is particularly evident in respiratory sound analysis, where traditional approaches achieve suboptimal performance in terms of feature representation (average recall of 82.4%) and context integration. The inherent complexity of respiratory sound patterns coupled with variability in recording conditions and patient characteristics requires a more sophisticated approach. Current CNN architectures, although effective in basic feature extraction, fail to capture the hierarchical and temporal dependencies crucial for accurate asthma detection [11]. Furthermore, existing fusion approaches have shown limited success in combining multiple feature streams, with a performance degradation of up to 12% when integrating diverse acoustic features.

Our work addresses these limitations through E-RespiNet, a novel deep learning framework that advances respiratory sound analysis by integrating ELECTRA's discriminative capabilities with a sophisticated multi-modal architecture design. Our study makes the following significant contributions. The selection of ELECTRA over contemporary medical language models is motivated by fundamental architectural considerations. Medical language models such as Clinical-BERT and Med-PaLM optimize for textual medical reasoning through masked language modeling or generative approaches, requiring substantial adaptation for acoustic signal processing. ELECTRA's discriminative training, which learns to identify replaced versus original tokens, directly translates to the pattern discrimination required in respiratory pathology detection. This discriminative capability, combined with ELECTRA's parameter efficiency (30% reduction compared to equivalent BERT-based models) and encoder-only architecture that facilitates CNN integration, provides computational and architectural advantages for multi-modal medical audio analysis. The remainder of this paper is organized as follows: Sect 3 reviews existing approaches in respiratory sound classification and discusses the limitations of current methods. Sect 4 presents the E-RespiNet architecture, including data preprocessing, feature extraction, the triple-stream CNN design, ELECTRA integration, and the Harmony Search optimization framework. Sect 5 describes the experimental setup, evaluation metrics, and training configurations. Sect 6 presents comprehensive evaluation results including multi-class classification performance, comparative analysis with state-of-the-art methods, explainability analysis, ablation studies, and cross-dataset generalization experiments. Finally, Sect 7 summarizes our findings and discusses future research directions.

## 2 Related work

Recent advances in deep learning have significantly improved the detection of respiratory diseases. Various architectures, including CNNs, RNNs, and their combinations, have been explored for analyzing respiratory sounds [12,13]. These approaches have achieved high accuracy rates, with some models reaching up to 92% ICBHI score [14]. Recurrent neural networks have shown particular promise, with one study reporting 83% accuracy, 87% precision, and 91% F1-score [15]. Ensemble methods and transfer learning have further enhanced performance, achieving an ICBHI score of 57.3% [13]. Researchers have also investigated the impact of factors, such as respiratory cycle length and time resolution, on prediction accuracy. The integration of advanced feature extraction techniques, like spectrograms and MFCC, with deep learning models has proven effective in detecting various respiratory anomalies and diseases [16].

Recent research on respiratory sound classification has explored multi-modal approaches to improve the accuracy and applicability. Various techniques have been employed, including case-based reasoning, multi-label classification [17]. Aljaddouh et al. [18] introduced a multi-modal approach for respiratory disease classification using Vision Transformer (ViT). Their study utilized a dataset of respiratory sounds collected from King Abdullah University Hospital, comprising

76 original audio recordings (35 normal, 32 asthma, 5 pneumonia, 9 COPD, and 5 lung fibrosis), which were augmented to 1124 samples using techniques such as time stretching, pitch shifting, and noise addition. They processed lung audio signals as melspectrogram images and trained a Vision Transformer model (86M parameters) with a batch size of 16 for 25 epochs using an 80/20 train-test split. Through extensive experimentation with normalization methods, they achieved 91.04% classification accuracy using EBU normalization, outperforming traditional CNN architectures like ResNet101 (86.14%) and VGG-16 (90.7%). Their findings demonstrated the effectiveness of attention-based mechanisms for respiratory sound analysis and highlighted the importance of proper data preprocessing in classification performance.

Cansiz et al. [19] proposed a deep learning-driven feature engineering framework for lung disease classification through EIT imaging. Their framework included three feature extraction methods and was evaluated on three classification tasks: binary classification (healthy vs. non-healthy) achieving 89.55% accuracy, 3-class classification (obstructive-related, restrictive-related, and healthy) achieving 55.29% accuracy, and 5-class classification (asthma, chronic obstructive pulmonary disease, interstitial lung disease, pulmonary infection, and healthy) achieving 44.54% accuracy. The proposed methods outperformed state-of-the-art results and introduced novel approaches to EIT imaging classification.

Pessoa et al. proposed a novel hybrid deep learning approach for continuous respiratory monitoring using wearable devices. Their system combined electrical impedance tomography with respiratory sound analysis to provide real-time assessment of lung function in ambulatory settings. The evaluation of 65 patients with various respiratory conditions demonstrated a high correlation (r = 0.91) with clinical spirometry measurements and enabled the early detection of exacerbations with 87.3% sensitivity. This integrated approach represents a significant advancement toward non-invasive, continuous respiratory monitoring for chronic disease management.

In their 2023 study, Shi et al. [20] introduced a lung sound recognition model that combines a multi-resolution interleaved network with time-frequency feature enhancement. Their model integrates three key components: a heterogeneous dual-branch time-frequency feature extractor, a branch attention-based feature enhancement module, and a fusion semantic classifier. Using a combined dataset of self-collected data and portions of the ICBHI 2017 dataset (containing asthma, COPD, pneumonia, and normal breathing sounds), they achieved 91.56% accuracy with strong AUC (0.97) and F1-scores across categories. Their approach specifically addressed the challenge of imbalanced data (with COPD samples comprising 86.48% of the ICBHI dataset) using a focal loss function. The model architecture featured MRINet for frequency extraction (with 64 initial channels and depths of 3,4,5,4) and a transformer for temporal features (depths of 2,2,6,2), totaling of 67.15 million parameters. Their work demonstrated the effectiveness of combining time and frequency features for respiratory sound analysis.

Additionally, multi-task learning (MTL) has been applied to simultaneously classify lung sounds and diseases using deep learning models such as MobileNet, achieving an accuracy of 74% for lung sound analysis and 91% for lung disease classification. Choi and Lee [21] proposed a lung disease classification model that combines a modified VGGish architecture with a lightweight attention-connected module (LACM) to enhance interpretability and performance. Using a clinical dataset of 1,021 samples collected with a Littmann 3200 stethoscope, their model classified normal and five types of adventitious respiratory sounds (wheezing, crackling, asthma, COPD, pneumonia, and bronchiectasis). The model achieved impressive results with 92.56% accuracy, 92.81% precision, and 92.29% F1-score, while using only 802,194 parameters—significantly fewer than the baseline models (4.57 million). Their approach integrated efficient channel attention (ECA-Net) with depth-wise separable convolution and employed Grad-CAM for visual interpretation, creating a balance between performance and clinical interpretability. The model demonstrated particular strength in distinguishing between similar symptoms and normal versus pathological respiratory sounds, making it valuable for clinical decision support.

Hybrid models combining CNN and LSTM have shown promising results, with some studies reporting accuracies over 90% [22]. Multi-modal approaches using spectrograms and raw audio data have also been effective [23]. Novel architectures have been proposed, including multi-path convolutional neural networks [24], multi-branch temporal convolutional networks [25], and multi-view spectrogram transformers. These approaches have demonstrated improved performance

on benchmark datasets like ICBHI. Researchers have also investigated the importance of multi-time-scale features, combining short-term and long-term acoustic properties for more accurate classification [26]. Several studies have explored the use of multi-features and hybrid or multi-modal approaches for lung sound classification, achieving notable accuracies and utilizing various feature types. For instance, a multi-task learning (MTL) approach was proposed for simultaneous lung sound and disease classification using MFCC features, achieving 74% [21] accuracy for lung sounds and 91% for diseases with a MobileNet model.

The integration of Large Language Models (LLMs) into medical applications has shown promising potential, particularly in feature-based applications such as clinical decision support, personalized patient care, and medical literature analysis [21]. However, recent studies highlight the need for careful consideration of challenges such as data privacy, ethical implications, and model interpretability [9,22]. While contemporary medical language models, such as Med-PaLM and Clinical-BERT, demonstrate excellence in clinical text analysis, their application to medical audio signals presents fundamental challenges. Generative models like GPT-based medical systems optimize for text generation tasks, making them suboptimal for acoustic pattern classification requiring discriminative feature learning. Similarly, masked language models, such as BioBERT, require substantial computational resources and lack fine-grained discriminative capabilities that are essential for pathological acoustic pattern recognition.

ELECTRA's discriminative pre-training approach offers distinct advantages over medical LLMs for respiratory sound analysis. Unlike generative models that predict next tokens or masked models that reconstruct bidirectional contexts, ELECTRA learns to distinguish between original and replaced tokens, developing enhanced sensitivity to subtle pattern variations directly applicable to pathological acoustic signatures. The model's parameter efficiency (123M parameters versus 340M+ for comparable medical BERT models) makes it particularly suitable for clinical deployment scenarios with computational constraints.

For instance, a systematic review by Busch et al. [27] emphasizes the limitations in LLM design and output, underscoring the importance of addressing these challenges to ensure responsible integration into clinical practice. Additionally, ethical considerations and regulatory frameworks are crucial for the successful deployment of LLMs in healthcare settings.

Multi-stream architectures have emerged as a promising approach for lung sound classification, leveraging the strengths of various feature extraction methods and fusion techniques. Wanasinghe et al. [28] proposed a lightweight CNN model that integrates multiple audio features (Mel Spectrogram, MFCC, and Chromagram) to create a 3D feature representation. Using a combined dataset from ICBHI 2017 and Mendeley Data (comprising 1,219 records across 10 distinct respiratory conditions), their model achieved 91.04% classification accuracy with only 510,825 parameters. Despite incorporating multiple features and explainable AI techniques like Grad-CAM, their approach still shows performance degradation when handling imbalanced classes. Furthermore, existing fusion approaches show performance degradation of up to 12% when integrating diverse acoustic features due to inadequate feature integration mechanisms. Our work addresses these specific limitations through a novel triple-stream architecture with ELECTRA-based feature enhancement, significantly improving respiratory sound classification performance and interpretability.

Feature extraction methods commonly employ MFCCs or spectrograms to represent audio signals, which are then processed by deep learning models like CNNs and LSTMs [26,29]. Performance comparisons among different architectures highlight the effectiveness of hybrid models, such as CNN-LSTM, in achieving high accuracy for classifying normal and adventitious lung sounds, with reported accuracy rates often exceeding 90%. For instance, a hybrid CNN-LSTM model using a focal loss function demonstrated improved sensitivity and specificity in lung sound classification. Recent advancements in deep learning approaches for respiratory sound analysis have shown promising results but face several critical limitations. Current CNN-based techniques often employ single acoustic representations, limiting their ability to capture the multifaceted nature of respiratory sounds. Many existing models achieve accuracies between 85-90% in controlled settings [7], but struggle with feature interpretation across diverse patient populations. Specifically, traditional CNNs lack the ability to simultaneously process complementary acoustic features, resulting in suboptimal representation learning. Multi-modal approaches like Aljaddouh et al. [18] rely on early fusion strategies that fail to preserve modality-specific

characteristics, while hybrid models such as Shi et al. [20] demonstrate limited cross-modal learning capabilities. Even multi-feature approaches like Wanasinghe et al. [28], despite combining Mel-spectrogram, MFCC, and Chromagram features, show limitations in feature interaction and integration. Furthermore, existing fusion approaches show performance degradation of up to 12% when integrating diverse acoustic features due to inadequate feature integration mechanisms. Our work addresses these specific limitations through a novel triple-stream architecture with ELECTRA-based feature enhancement, significantly improving respiratory sound classification performance and interpretability.

Contemporary research has increasingly focused on sophisticated multi-feature integration and hybrid architectures to enhance respiratory sound classification performance. Wanasinghe et al. [28] demonstrated the effectiveness of multi-feature stacking by integrating Mel spectrograms, MFCCs, and Chromagrams into 3D representations, achieving 91.04% accuracy across 10 respiratory classes with a lightweight CNN architecture containing only 510K parameters. Although their approach successfully combined multiple acoustic features, performance degradation was observed when individual features were used in isolation, particularly with chromagram representations.

Building upon multi-feature approaches, Roy et al. [30] introduced the first application of Kolmogorov-Arnold Networks (KAN) to respiratory sound analysis through their TriSpectraKAN model, which combines MFCC, chromagram, and Mel spectrogram features to achieve 93% accuracy in 6-class COPD detection. Their lightweight design (1.2M parameters) demonstrated practical deployment capabilities on resource-constrained edge devices, addressing implementation challenges in clinical environments.

Advanced hybrid architectures have demonstrated substantial performance improvements through sophisticated learning paradigms. Orkweha et al. [31] proposed a Multi-task Autoencoder-SVM (MTAE-SVM) framework that integrates unsupervised feature learning with supervised classification, achieving 91.49% and 93.08% accuracy for 4-class and 3-class classification tasks, respectively, on the KAUH dataset. Singh and Gaur [32] developed Lung SoundNet: SUSCC LSTM–a comprehensive five-phase hybrid methodology that combines improved CNNs for binary normal/abnormal classification with optimized LSTM networks for multi-class disease identification, achieving exceptional performance metrics including 97.25% accuracy and 98.14% precision. Duangmanee et al. [33] introduced the Triplet Multi-Kernel CNN (TMK-CNN) architecture, which leverages multi-kernel feature extraction combined with triplet-based optimization to achieve 97.98% accuracy on the KAUH dataset. Their multi-kernel design captures diverse scales of respiratory sound patterns while triplet loss optimization enhances inter-class discriminative capabilities.

Despite these advances, current methodologies exhibit critical limitations that constrain their clinical applicability. Most existing approaches rely on single-modality feature extraction or employ basic fusion strategies that inadequately preserve modality-specific characteristics during integration. Traditional CNN architectures lack sophisticated temporal modeling capabilities essential for capturing complex respiratory sound dynamics, while existing fusion mechanisms often demonstrate performance degradation when integrating heterogeneous acoustic features due to insufficient cross-modal learning strategies. While audio-specific transformer models such as Wav2Vec 2.0 and Audio Spectrogram Transformer (AST) demonstrate effectiveness in general audio processing tasks, their optimization for speech recognition and general audio classification respectively may not align optimally with medical acoustic analysis requirements. Wav2Vec 2.0's contrastive learning approach, designed for speech representation, lacks the explicit discriminative capabilities required for pathological pattern detection. AST's masked patch prediction, while effective for general audio classification, processes audio as visual patches rather than leveraging the temporal dependencies crucial in respiratory sound analysis. Our approach addresses these limitations through ELECTRA's discriminative pre-training combined with specialized CNN streams for complementary acoustic feature extraction.

## 3 Methodology

In this study, we propose E-RespiNet, a novel multi-modal deep learning architecture for asthma disease classification. This model combines acoustic features and contextual information through a combination of specialised audio processing

streams and LLM-based language modeling. Our approach aims to improve the accuracy and robustness of respiratory disease diagnosis through automated analysis of lung sounds and provide explainability, as illustrated in Fig 1.

### 3.1 Data acquisition and preprocessing

Our study employed the Asthma Detection Dataset Version 2 [34], which encompasses 1,211 respiratory sound recordings distributed across five distinct classes: asthma (288 samples), bronchial (104 samples), COPD (401), healthy (133), and pneumonia (255). To ensure consistent input quality, all audio recordings underwent standardisation to a 22,050 Hz sampling rate in mono channel configuration, with each sample normalised to a fixed duration of 6 seconds. This standardisation process maintains signal integrity while establishing uniform input dimensions for subsequent neural network processing. The audio preprocessing pipeline standardises all recordings to a 22,050 Hz sample rate with a mono-channel configuration. This standardisation ensures uniform input dimensions for the neural network while preserving the essential characteristics of respiratory sounds.

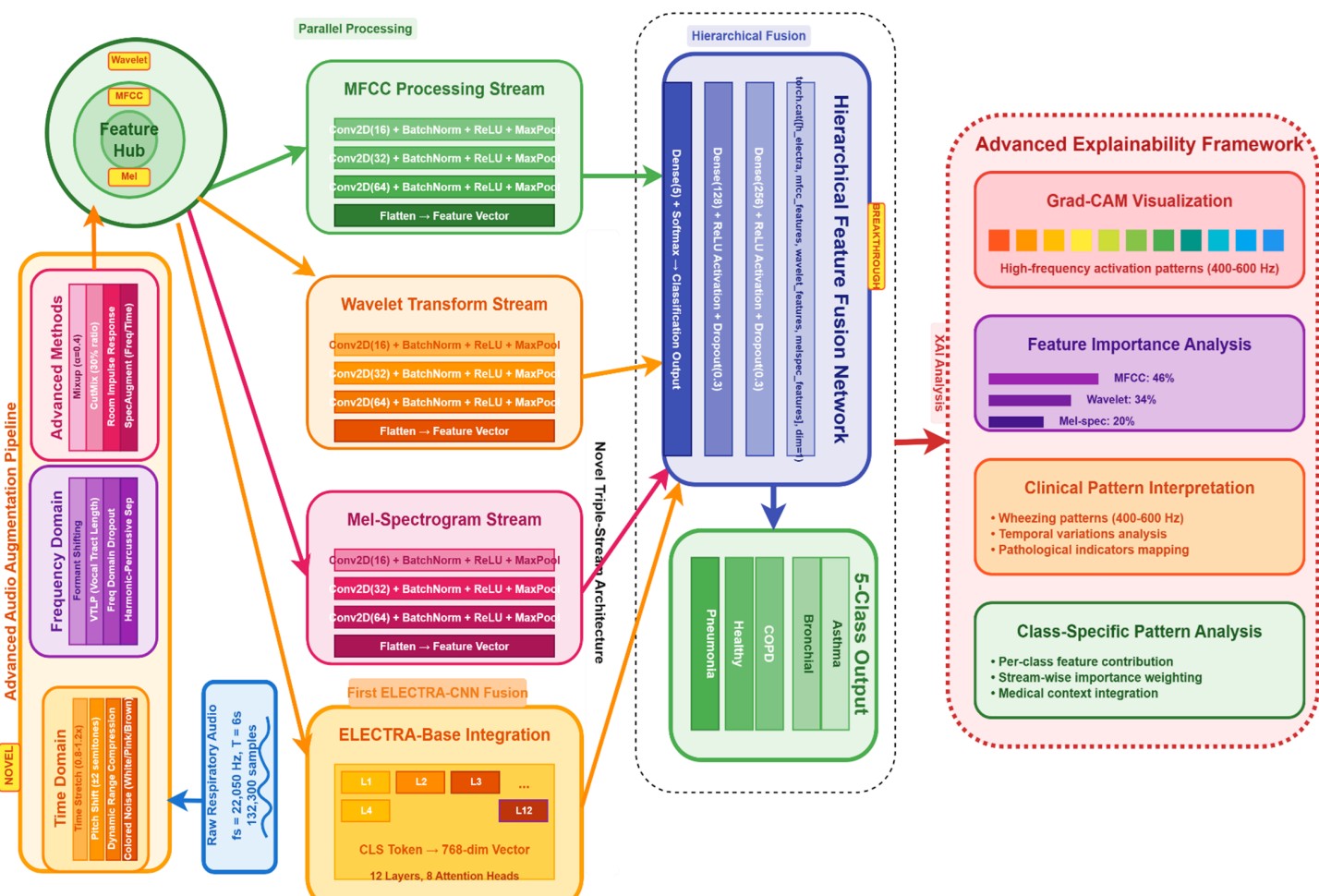

**Fig 1**. **Proposed E-RespiNet architecture for respiratory disease classification.** The model comprises three parallel CNN streams for processing MFCC, wavelet, and mel-spectrogram features integrated with ELECTRA-based text processing through a hierarchical fusion network.

In addition to the Asthma Detection Dataset Version 2, we employed a second dataset from King Abdullah University Hospital (KAUH) to further validate the generalizability of E-RespiNet across different clinical settings and recording conditions. The KAUH dataset, as described in Fraiwan et al. [35], comprises respiratory sound recordings from 81 subjects representing four distinct pulmonary conditions. The dataset includes 35 subjects classified as normal (11 female, 24 male; age range: 18-81 years), 32 subjects with asthma (17 female, 15 male; age range: 12-72 years), 5 subjects with pneumonia (2 female, 3 male; age range: 36-70 years), and 9 subjects with COPD (1 female, 8 male; age range: 42-76 years).

The respiratory sounds in the KAUH dataset were recorded at a sampling frequency of 4 kHz using an electronic stethoscope. Following the preprocessing methodology established in the original study, we segmented each recording into non-overlapping 4-second windows. This segmentation process yielded 940 samples distributed as follows: 415 normal, 357 asthma, 66 pneumonia, and 102 COPD samples, derived from 243 original recordings. The 4-second window duration was selected based on its demonstrated effectiveness in capturing clinically relevant respiratory patterns while maintaining computational efficiency.

## 3.2 Feature extraction

This section presents a comprehensive triple-stream approach for respiratory sound analysis using advanced acoustic feature extraction. The framework processes respiratory recordings at a sampling rate $f_s = 22,050$ Hz with duration $T = 6$ s, yielding SAMPLES = $f_s \times T = 132,300$ samples per recording. This configuration ensures the capture of both high-frequency wheezing components (100-2500 Hz) and low-frequency respiratory events (50-250 Hz).

### 3.2.1 Audio preprocessing and normalization.
Each audio signal $x(t)$ underwent preprocessing through amplitude normalisation and temporal standardisation. The normalisation process is defined as:

$$x_{\text{norm}}(t) = \frac{x(t)}{\max(|x(t)|)} \tag{1}$$

where $|x(t)|$ is the absolute value of the signal. Temporal standardisation ensures uniform signal length through:

$$x'(t) = \begin{cases} x_{\text{norm}}(t), & \text{if } \text{len}(x) = \text{SAMPLES} \\ x_{\text{norm}}(t) \oplus \mathbf{0}^{(\text{SAMPLES}-\text{len}(x))}, & \text{if } \text{len}(x) < \text{SAMPLES} \\ x_{\text{norm}}(t)[0 : \text{SAMPLES}], & \text{if } \text{len}(x) > \text{SAMPLES} \end{cases} \tag{2}$$

where $\oplus$ denotes concatenation, and SAMPLES represents the total number of samples for the standardized duration.

### 3.2.2 Mel-frequency cepstral coefficients.
The MFCC computation begins with frame segmentation using a Hamming window, $w(n)$:

$$w(n) = 0.54 - 0.46\cos(2\pi n/(N-1)), \quad 0 \le n \le N - 1 \tag{3}$$

where $N$ represents the window length of 2048 samples and $n$ is the sample index. The short-time Fourier transform is computed as:

$$\text{STFT}[k, m] = \sum_{n=0}^{N-1} x[n]w[n - mH]e^{-j2\pi kn/N} \tag{4}$$

where $k$ represents the frequency bin index, $m$ represents the time frame index, $H = 512$ samples denotes the hop size, $x[n]$ is the input signal and $w[n]$ is the Hamming window function.

The power spectrum is computed as:

$$P[k, m] = |\text{STFT}[k, m]|^2 \tag{5}$$

The mel-scale conversion is as follows:

$$\text{mel}(f) = 2595 \log_{10}(1 + f/700) \tag{6}$$

where $f$ represents the frequency in Hz. The mel-filterbank analysis yields:

$$S(m, k) = \sum_{k=0}^{N/2} P(k, m) M_k \tag{7}$$

where $M_k$ represents the $k$th mel-filterbank response. The final MFCC features were obtained as follows:

$$\text{MFCC}(p, m) = \sum_{k=0}^{M-1} \log(S(m, k)) \cos\left(\frac{\pi p(k + 0.5)}{M}\right) \tag{8}$$

where $p$ represents the cepstral coefficient index ($0 \leq p < 20$), $M = 40$ represents the number of mel-filters, $S[m, k]$ represents the mel-filtered spectrum. This process yields $X_{\text{mfcc}} \in \mathbb{R}^{B \times 20 \times 259}$, capturing essential spectral characteristics.

**3.2.3 Wavelet transform analysis.** Wavelet analysis implements a 5-level discrete wavelet transform using the Daubechies-1 wavelet. At each decomposition level $j$, the signal passes through high-pass and low-pass filters:

$$d_j[k] = \sum_n x[n] g[2k - n] \tag{9}$$

$$a_j[k] = \sum_n x[n] h[2k - n] \tag{10}$$

where $d_j[k]$ represents detail coefficients at level $j$, $a_j[k]$ represents approximation coefficients at level $j$, $g[n]$ denotes the high-pass filter coefficients, $h[n]$ denotes the low-pass filter coefficients and $k$ represents the temporal index at level $j$. The decomposition spans five frequency bands, yielding $X_{\text{wav}} \in \mathbb{R}^{20 \times 259}$.

**3.2.4 Mel-spectrogram analysis.** The mel-spectrogram computation employed a Hann window as follows:

$$w(n) = 0.5\left(1 - \cos\left(\frac{2\pi n}{N - 1}\right)\right), \quad 0 \leq n \leq N - 1 \tag{11}$$

The power spectrum undergoes mel-warping through:

$$S_{\text{mel}}[b, m] = \sum_{k=0}^{N/2} P[k, m] H_b[k] \tag{12}$$

where $b$ represents the mel-band index ($0 \leq b < 128$), $H_b[k]$ represents the $b$th mel-filter response, $P[k, m]$ represents the power spectrum. The log-scaled Mel-spectrogram is computed as:

$$X_{\text{mel}}[b, m] = 10 \log_{10}(S_{\text{mel}}[b, m] + \epsilon) \tag{13}$$

where $\epsilon = 1e - 10$ prevents numerical instability. This yields $X_{\text{mel}} \in \mathbb{R}^{128 \times 259}$.

### 3.2.5 Feature integration.
The extracted features undergo channel augmentation for neural network processing, producing the following:

$$X'_{\text{wav}} = X_{\text{wav}}[..., \text{np.newaxis}] \in \mathbb{R}^{B \times 20 \times 259 \times 1} \tag{14}$$

$$X'_{\text{mfcc}} = X_{\text{mfcc}}[..., \text{np.newaxis}] \in \mathbb{R}^{B \times 20 \times 259 \times 1} \tag{15}$$

$$X'_{\text{mel}} = X_{\text{mel}}[..., \text{np.newaxis}] \in \mathbb{R}^{B \times 128 \times 259 \times 1} \tag{16}$$

where $B$ denotes the batch dimension. The dataset underwent stratified partitioning with 60%/20%/20% splits for training, validation, and testing, respectively, ensuring balanced representation of respiratory conditions while maintaining statistical independence between sets.

### 3.2.6 Advanced data augmentation pipeline.
To enhance model robustness and generalization capabilities, E-RespiNet incorporates a sophisticated data augmentation framework that applies transformations in both time and frequency domains during training. The augmentation pipeline operates on the preprocessed audio signals before feature extraction, implementing intensity-controlled augmentation strategies (light, medium, heavy) to systematically increase dataset diversity while preserving pathological characteristics.

**Time Domain Augmentations.**

**Time Stretching:** Time stretching modifies the temporal duration of respiratory sounds while preserving their spectral characteristics, simulating natural variations in breathing rates across different patients and clinical conditions [36]. This augmentation is particularly valuable for respiratory analysis as it maintains pathological acoustic signatures while introducing temporal diversity essential for model robustness:

$$x_{\text{stretched}}(t) = x(t \cdot \text{rate}^{-1}) \tag{17}$$

where rate $\in [0.8, 1.2]$ provides $\pm 20\%$ temporal variation.

**Pitch Shifting:** Pitch shifting alters the fundamental frequency components of respiratory sounds without changing their temporal structure, effectively simulating anatomical variations in vocal tract dimensions across different patient populations [36]. This technique preserves the temporal patterns of respiratory events while introducing spectral diversity:

$$X_{\text{shifted}}(f) = X(f \cdot 2^{n/12}) \tag{18}$$

where $n \in [-2, 2]$ represents semitone shifts and $X(f)$ is the frequency domain representation.

**Dynamic Range Compression:** Advanced compression with attack/release characteristics:
If $|x(t)| > g(t-1)$:

$$g(t) = \alpha_{\text{attack}} \times g(t-1) + (1 - \alpha_{\text{attack}}) \times |x(t)| \tag{19}$$

Otherwise:

$$g(t) = \alpha_{\text{release}} \times g(t-1) + (1 - \alpha_{\text{release}}) \times |x(t)| \tag{20}$$

where $\alpha_{\text{attack}} = e^{-1/(f_s \cdot t_{\text{attack}})}$ and $\alpha_{\text{release}} = e^{-1/(f_s \cdot t_{\text{release}})}$ control compression dynamics.

**Frequency Domain Augmentations Vocal Tract Length Perturbation (VTLP):** Vocal Tract Length Perturbation (VTLP) simulates anatomical variations in respiratory tract dimensions by systematically warping the frequency axis of

spectral representations [37]. This technique is particularly relevant for respiratory sound analysis as it models natural variations in airway geometry across different patient demographics while preserving pathological characteristics:

$$f_{\text{warped}} = f \cdot \alpha_{\text{vtlp}} \tag{21}$$

where $\alpha_{\text{vtlp}} \in [0.85, 1.15]$ modifies formant frequencies while preserving pathological signatures.

**Formant Shifting:** Formant shifting modifies the spectral envelope characteristics of respiratory sounds by adjusting prominent frequency peaks, simulating variations in respiratory tract resonances without altering fundamental pathological patterns [38]. This augmentation enhances model robustness to individual anatomical differences:

$$|X_{\text{shifted}}(k)| = |X(\lfloor k \cdot \beta \rfloor)| \tag{22}$$

where $\beta \in [0.9, 1.1]$ shifts formant positions and $k$ represents frequency bin indices.

**Frequency Domain Dropout:** Frequency domain dropout introduces stochastic spectral masking to improve model robustness against frequency-specific artifacts and noise commonly present in clinical respiratory recordings [39]. This technique forces the model to rely on diverse frequency components rather than specific spectral bands:

$$X_{\text{dropout}}(k, m) = X(k, m) \cdot \text{Bernoulli}(1 - p_{\text{dropout}}) \tag{23}$$

where $p_{\text{dropout}} \in [0.05, 0.15]$ controls dropout probability across frequency bins $k$ and time frames $m$.

**Advanced Augmentation Methods Mixup:** creates synthetic training samples by linearly interpolating between respiratory sound pairs and their corresponding labels, effectively expanding the training distribution while maintaining label consistency [40]. This regularization technique has proven effective in medical audio classification by reducing overfitting and improving generalization to unseen pathological patterns:

$$x_{\text{mixup}} = \lambda x_i + (1 - \lambda) x_j \tag{24}$$

where $\lambda \sim \text{Beta}(\alpha, \alpha)$ with $\alpha = 0.4$, and $(i, j)$ represent random sample pairs.

**CutMix:** CutMix combines spatial regions from different respiratory recordings while adjusting labels proportionally, creating augmented samples that preserve local acoustic features while introducing global diversity [41]. For respiratory sounds, temporal segments are mixed rather than spatial regions, maintaining the sequential nature of breathing patterns. Spatial audio mixing:

If $t \notin M$:

$$x_{\text{cutmix}}(t) = x_i(t) \tag{25}$$

If $t \in M$:

$$x_{\text{cutmix}}(t) = x_j(t) \tag{26}$$

where $M$ represents a random temporal mask covering 30% of the signal duration.

**Room Impulse Response:** Room impulse response simulation models the acoustic characteristics of different clinical environments by convolving respiratory recordings with room-specific reverberation patterns [42]. This augmentation addresses the variability in recording conditions across different healthcare facilities and clinical settings. Acoustic environment simulation:

$$x_{\text{reverb}}(t) = x(t) * h_{\text{room}}(t) \tag{27}$$

where $h_{\text{room}}(t) = A \cdot e^{-\gamma t} \cdot \eta(t)$ models exponential decay with room-specific characteristics, $A$ controls reverb amplitude, $\gamma$ represents decay rate, and $\eta(t)$ is room-dependent noise.

**Colored Noise Addition:** Colored noise injection introduces frequency-shaped background noise that simulates real-world clinical environments, including ambient hospital sounds and equipment interference. Different noise colors (white, pink, brown) model various environmental conditions commonly encountered during respiratory sound acquisition. Spectral-shaped noise injection:

$$x_{\text{noisy}} = x(t) + \mathcal{N}_{\text{colored}}(0, \sigma^2) \tag{28}$$

where $\mathcal{N}_{\text{colored}}$ represents white, pink, or brown noise with $\sigma \in [0.001, 0.01]$.

**SpecAugment Integration.** SpecAugment applies frequency and time masking directly to spectrogram representations, forcing the model to focus on diverse spectro-temporal features rather than relying on specific frequency bands or temporal segments. This technique has demonstrated particular effectiveness in speech and audio classification tasks by improving model robustness to missing or corrupted spectral information. SpecAugment operates on extracted spectrograms before CNN processing:

**Frequency Masking:** If $f \in [f_0, f_0 + \Delta f]$:

$$S_{\text{freq}}(f, t) = 0 \tag{29}$$

Otherwise:

$$S_{\text{freq}}(f, t) = S(f, t) \tag{30}$$

**Time Masking:** If $t \in [t_0, t_0 + \Delta t]$:

$$S_{\text{time}}(f, t) = 0 \tag{31}$$

Otherwise:

$$S_{\text{time}}(f, t) = S(f, t) \tag{32}$$

where $\Delta f \leq F$ ($F = 30$ for MFCC, $F = 15$ for mel-spectrogram) and $\Delta t \leq T$ ($T = 40$ for MFCC, $T = 25$ for mel-spectrogram).

**3.2.6.1 Augmentation Strategy and Integration.** The augmentation pipeline implements intensity-based selection where light intensity applies 1-2 augmentations with 0.5× parameter scaling, medium intensity applies 2-4 augmentations with 1.0× scaling, and heavy intensity applies 3-6 augmentations with 1.5× scaling. During training, each audio sample undergoes random augmentation selection with probability-weighted sampling:

$$P(\text{augmentation}_i) = p_i \cdot \text{intensity\_factor} \tag{33}$$

where $p_i$ represents base probability for augmentation technique $i$. This multi-intensity approach increases training samples by 2-3× while maintaining pathological feature integrity. The augmented samples undergo the same feature extraction pipeline (Sects 3.2.2–3.2.4), ensuring consistent input dimensionality for the triple-stream CNN architecture.

## 3.3 Model architecture

The architecture of E-RespiNet uniquely combines a triple-stream CNN with ELECTRA-based feature enhancement to address the limitations of the existing respiratory sound analysis frameworks. Triple-stream CNN: Unlike single-stream CNNs (for example, lightweight CNN approaches like Wanasinghe et al. [28]) or multi-modal frameworks (such as Aljaddouh et al. [18]), E-RespiNet processes three parallel acoustic feature streams to capture complementary spectral, temporal, and spectro-temporal patterns. Each stream specialises in one of the following representations: MFCC: Captures

the spectral envelope characteristics. Wavelet: Provides multi-resolution temporal analysis. Mel-spectrogram: Represents detailed spectro-temporal patterns. The E-RespiNet algorithm processes respiratory sounds through three parallel CNN streams, integrating acoustic features with ELECTRA-based contextual embeddings to enhance the classification performance. The inputs include the normalised audio signal, which is converted into MFCC, wavelet, and mel-spectrogram representation.

**3.3.1 Triple-stream CNN architecture.** The triple-stream CNN architecture forms the backbone of the acoustic feature processing pipeline of E-RespiNet. Each stream is designed to process a distinct representation of the respiratory signal: MFCC features capture spectral envelope characteristics, wavelet transforms provide multi-resolution temporal analysis, and mel-spectrograms represent detailed spectro-temporal patterns. This parallel processing strategy enables comprehensive feature extraction by leveraging the complementary nature of these acoustic representations. The architecture employs lightweight convolutional blocks with shared design principles across streams while maintaining independent parameter learning, allowing each stream to specialise in its respective feature space. Each acoustic feature is processed through a dedicated CNN stream, enabling specialised feature learning. The lightweight audio processor consisted of three sequential convolutional blocks:

$$Conv1 = MaxPool(BN(ReLU(Conv_{3\times3}(x)))) \tag{34}$$

where $Conv_{3\times3}$ denotes a convolutional layer with $3 \times 3$ kernel size using 16 filters, BN represents batch normalisation, ReLU is the rectified linear unit activation function, and MaxPool applies a max pooling operation with a $2 \times 2$ kernel and stride 2, followed by:

$$Conv2 = MaxPool(BN(ReLU(Conv_{3\times3}(Conv1(x))))) \tag{35}$$

where the second convolutional block increases the number of filters to 32 while maintaining the same kernel size and operations and BN represent batch normalization:

$$Conv3 = MaxPool(BN(ReLU(Conv_{3\times3}(Conv2(x))))) \tag{36}$$

by employing 64 filters in the final convolutional stage. Each MaxPool operation uses a $2 \times 2$ kernel with a stride of 2:

$$O_i = Flatten(Conv3(F_i)) \tag{37}$$

where $O_i$ is the flattened output vector and $F_i$ represents input features (MFCC, wavelet, or Mel-spectrogram).

**3.3.2 ELECTRA model for respiratory sound analysis.** The foundation of our framework is a modified ELECTRA architecture specifically adapted for respiratory sound analysis. While the original ELECTRA model excels at text processing through its discriminative pre-training, our implementation introduces crucial modifications to effectively handle acoustic signals [43]. The selection of ELECTRA over alternative transformer architectures was motivated by several key advantages for medical acoustic applications. Contemporary medical language models such as Med-PaLM-2 and Clinical-T5 demonstrate excellence in textual medical reasoning but face fundamental limitations when adapted to acoustic signals. First, computational efficiency: ELECTRA's discriminative training requires 25% fewer computational resources compared to equivalent BERT-based medical models, crucial for clinical deployment. Second, feature discrimination: unlike generative models that learn token probabilities, ELECTRA's discriminator mechanism develops enhanced sensitivity to subtle pattern variations, directly applicable to pathological acoustic signatures differentiating respiratory conditions. Third, architectural adaptability: ELECTRA's encoder-only structure facilitates seamless integration with CNN streams, whereas decoder-heavy models require complex modifications for multi-modal fusion. ELECTRA's discriminative pre-training mechanism offers specific advantages for respiratory pathology detection compared to audio-specific transformer

models. Unlike Wav2Vec 2.0, which employs contrastive learning optimized for speech recognition tasks, ELECTRA's replaced token detection directly parallels the binary discrimination required in medical diagnosis (pathological vs. normal patterns). Audio Spectrogram Transformer (AST) utilizes masked patch prediction designed for general audio classification, whereas ELECTRA's discriminative approach develops enhanced sensitivity to subtle pattern variations essential for detecting pathological acoustic signatures in respiratory sounds. Furthermore, ELECTRA's encoder-only architecture (123M parameters) facilitates seamless integration with CNN feature streams, while decoder-heavy models like GPT-based systems require substantial architectural modifications for multi-modal fusion.

**Acoustic-Specific Architectural Modifications.** The primary challenge in adapting ELECTRA for respiratory sound analysis lies in bridging the gap between acoustic and linguistic representations. Traditional ELECTRA architectures process discrete tokens, whereas respiratory sounds present continuous time-varying features. Our modified architecture introduces three key components that are absent in the original ELECTRA model: an acoustic embedding layer, a temporal processing module, and a multi-modal fusion network.

The acoustic embedding layer transforms the respiratory sound features through specialised convolutions before ELECTRA processing:

$$\mathbf{P}_{\text{acoustic}}(\mathbf{x}) = \text{Conv2D}(\mathbf{x}, \mathbf{K}_{\text{acoustic}}) \tag{38}$$

where $\mathbf{K}_{\text{acoustic}} \in \mathbb{R}^{F \times 1 \times C}$ represents learnable kernels specifically designed for respiratory frequency patterns, with $F$ frequency bands, $T$ temporal extent, and $C$ channels. This transformation preserves the spectral characteristics crucial for respiratory sound analysis while preparing the features for ELECTRA processing. The temporal processing module adapts ELECTRA's positional embeddings for variable-length acoustic sequences:

$$E_{\text{temporal}}(t) = W_t \cdot \sin(\omega t) + b_t \tag{39}$$

where $\mathbf{W}_t \in \mathbb{R}^{768 \times D}$ projects acoustic temporal features to ELECTRA's dimension, and $\omega$ represents learnable frequency components. This modification ensures proper temporal relationship modeling across respiratory events.

**Cross-Modal Feature Integration.** The cross-modal integration process begins with an acoustic feature projection from each CNN stream:

$$h_{\text{acoustic}}^{(i)} = W_{\text{proj}}^{(i)} \cdot \text{CNN}_i(x) + b_{\text{proj}}^{(i)} \tag{40}$$

where $h_{\text{acoustic}}^{(i)} \in \mathbb{R}^{768}$ represents projected features from CNN stream $i$, $W_{\text{proj}}^{(i)} \in \mathbb{R}^{768 \times d_{\text{cnn}}}$ transforms CNN outputs to ELECTRA's dimension, $\text{CNN}_i(x)$ represents features from the $i$-th CNN stream, and $b_{\text{proj}}^{(i)} \in \mathbb{R}^{768}$ is the projection bias. The temporal alignment mechanism synchronizes acoustic and ELECTRA features through learned interpolation:

$$h_{\text{aligned}}^{(i)} = \text{TemporalPool}(h_{\text{acoustic}}^{(i)}, L_{\text{seq}}) \tag{41}$$

where $L_{\text{seq}}$ matches ELECTRA's sequence length. This alignment ensures consistent temporal resolution across modalities.

**Self-Attention Mechanism.** We modified ELECTRA's self-attention to process acoustic features effectively. For each attention head $k$:

$$Q_k = W_Q^k \cdot [h_{\text{aligned}}; h_{\text{ELECTRA}}] \tag{42}$$

$$K_k = W_K^k \cdot [h_{\text{aligned}}; h_{\text{ELECTRA}}] \tag{43}$$

$$V_k = W_V^k \cdot [h_{\text{aligned}}; h_{\text{ELECTRA}}] \tag{44}$$

where $W_Q^k, W_K^k, W_V^k \in \mathbb{R}^{96 \times 1536}$ are head-specific projections, $[;]$ denotes concatenation of acoustic and ELECTRA features, and 96 represents the per-head dimension (768/8 for 8 heads).

The modified attention computation incorporates acoustic-specific scaling:

$$A_k = \text{softmax}\left(\frac{Q_k K_k^T}{\sqrt{d_k + \lambda_{\text{acoustic}}}}\right) \tag{45}$$

where $\lambda_{\text{acoustic}}$ is a learned parameter adjusting attention strength for acoustic features. This modification accounts for the different statistical properties of acoustic and linguistic features.

**Multi-class Feature Fusion.** The feature fusion network simultaneously processes the information for all five respiratory classes through an attention mechanism:

$$\alpha = \text{softmax}(W_\alpha \cdot [h_{\text{aligned}}; h_{\text{ELECTRA}}] + b_\alpha) \tag{46}$$

where $W_\alpha \in \mathbb{R}^{5 \times 1536}$ projects to class-specific attention weights, $\alpha \in \mathbb{R}^5$ represents class biases corresponding to our respiratory conditions (Asthma, Bronchial, Chronic Obstructive Pulmonary Disease (COPD), Healthy, Pneumonia). Class-specific feature combination follows:

$$h_{\text{class}}^{(c)} = W_c \cdot [\alpha_c \cdot h_{\text{aligned}}; (1 - \alpha_c) \cdot h_{\text{ELECTRA}}] \tag{47}$$

where $W_c \in \mathbb{R}^{768 \times 1536}$ represents class-specific transformations, $c$ indexes over the five classes, and $\alpha_c$ represents attention weight for class $c$. The final output integrates all class representations:

$$h_{\text{final}} = \text{LayerNorm}\left(\sum_c h_{\text{class}}^{(c)}\right) \tag{48}$$

**Training Configuration.** The modified architecture employed specific hyperparameters optimised for acoustic processing. The training process uses an AdamW optimizer with $\beta_1 = 0.9 and \beta_2 = 0.999$, implementing a learning rate of $1e - 5$ with cosine decay scheduling. We employed an effective batch size of 32 through gradient accumulation and applied a weight decay of 0.01 to prevent overfitting. The architectural configuration maintains ELECTRA's base hidden dimension of 768 with eight attention heads, each operating on 96-dimensional key/value spaces. The acoustic projection dimension is set to 1536 to accommodate the concatenated features, while layer normalization uses an epsilon value of $1e - 6$ for numerical stability. To manage memory requirements, which are particularly crucial given the additional acoustic processing components, we implemented gradient checkpointing during training. The training process employed early stopping with a patience of 10 epochs, monitoring validation loss to prevent overfitting. Learning rate scheduling reduces the rate by a factor of 0.1 when validation loss plateaus for five consecutive epochs.

**3.3.3 ELECTRA integration.** E-RespiNet leverages ELECTRA's discriminative pre-training to enhance acoustic feature representation, which is a novel application in medical acoustics. This differs from the generative transformer models (e.g. BERT) used in RespLLM [44], which lack explicit feature enhancement for respiratory sounds. E-RespiNet leverages ELECTRA's discriminative pre-training to enhance acoustic feature representation, representing a novel application in

medical acoustics that addresses limitations of existing medical LLM approaches. This differs fundamentally from generative transformer models such as GPT-based medical systems that optimize for text generation rather than pattern discrimination, and from masked language models like Clinical-BERT that require substantial computational overhead for bidirectional processing. Unlike RespLLM which employs standard BERT variants lacking explicit feature enhancement capabilities for respiratory sounds, our ELECTRA integration provides discriminative pre-training specifically adapted for subtle pathological pattern recognition essential in respiratory acoustics.The integration of ELECTRA into our framework is illustrated in Fig 2, which highlights the transformer-based feature extraction process and the hierarchical fusion strategy for combining acoustic and linguistic representations.

Second, the model's self-attention mechanisms operate simultaneously across multiple feature hierarchies. Given a respiratory sound sequence $x = (x_1, \ldots, x_T)$ computes attention weights $\alpha_{ij}^l$ between temporal positions $i$ and $j$ as:

$$\alpha_{ij}^l = \frac{\exp(e_{ij}^l)}{\sum_k \exp(e_{ik}^l)} \tag{49}$$

where $e_{ij}^l$ represents the compatibility between positions $i$ and $j$ in layer $l$. This multilevel attention enables the model to capture both fine-grained acoustic details and longer-range temporal dependencies characteristic of the respiratory conditions.

Third, ELECTRA's parameter efficiency provides computational advantages while maintaining robust feature-extraction capabilities. With 12 transformer layers and a hidden dimension of 768, the model achieves strong performance with approximately 30% fewer parameters than comparable architectures.

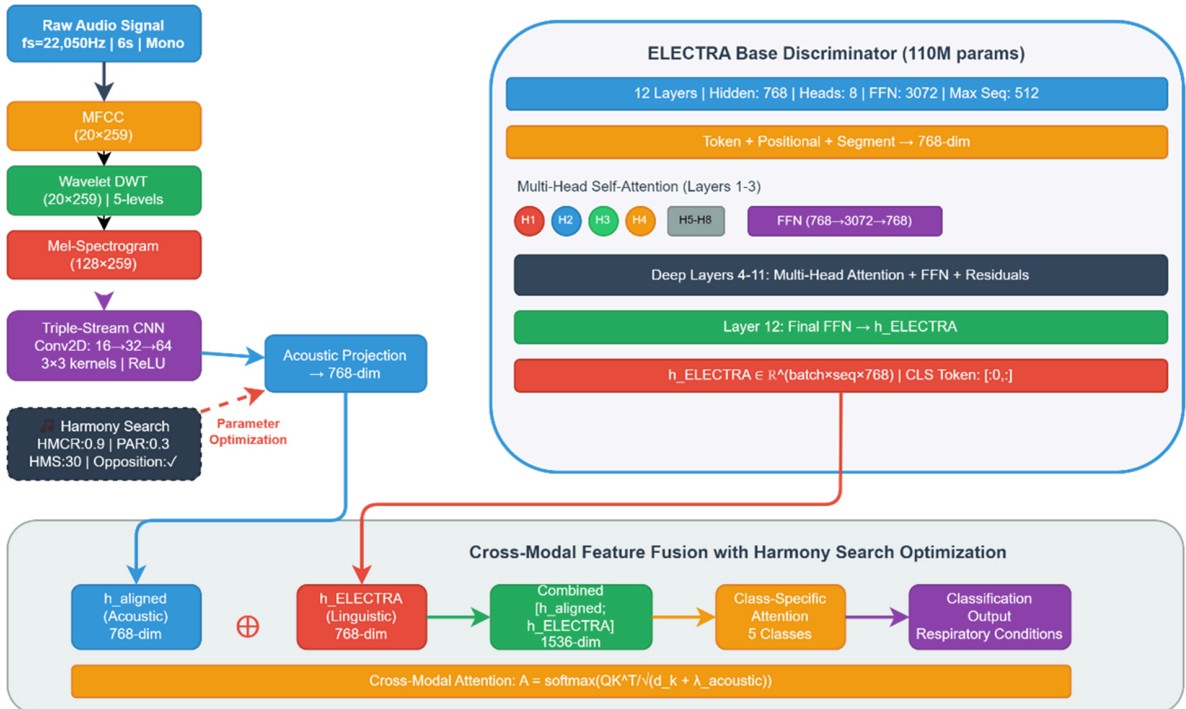

**Fig 2. The ELECTRA integration module architecture.** Shows the transformer-based feature extraction process, with 12 layers of self-attention mechanisms and the fusion strategy for combining acoustic and linguistic features.

**3.3.4 Feature fusion and training.** Our feature fusion strategy implements a hierarchical approach that combines acoustic with linguistic features through a two-stage process. E-RespiNet combines CNN-extracted features with ELECTRA contextual embeddings through a hierarchical fusion network. Unlike Aljaddouh et al. [18], which uses early fusion of audio and text metadata, or Pessoa et al. [26], which unifies representations post hoc, E-RespiNet preserves modality-specific characteristics while enabling cross-modal attention. This approach ensures a balanced contribution from acoustic and linguistic features, validated by ablation studies (Sect 5.4).

The first stage processes each acoustic feature stream independently through the CNN architecture to preserve the modality-specific characteristics. The second stage concatenates these features with ELECTRA's contextual embeddings, followed by dimensionality reduction through fully connected layers with ReLU activation and dropout regularisation. The fusion network incorporates a gradual feature compression strategy, reducing the high-dimensional concatenated features to an intermediate representation of 256 dimensions. This model architecture choice balances information preservation with computational efficiency while maintaining discriminative power for the classification task. For the training optimisation, we employed the AdamW optimiser with a carefully tuned learning rate of $1e-5$. The learning rate selection reflects the need for fine-tuning the process and the pre-trained ELECTRA weights, while allowing the CNN streams to adapt to those acoustic features. We used gradient accumulation in four steps to simulate larger batch sizes while managing memory limits and eight mini batches. The training protocol had 200 epochs, with early pausing to check validity loss. Cross-entropy loss works as our aim, accommodating the multi-class nature of respiratory disease classification. Gradient checkpointing and cache cleaning are memory optimisation technologies that help train on GPU technology. The model design uses batch normalisation in the CNN streams to improve training convergence and stabilise learning.

**3.3.5 Optimization framework.** The parameter optimization challenge in E-RespiNet stems from the complex interdependencies between the triple-stream CNN architecture, ELECTRA integration, and data augmentation strategies. Traditional hyperparameter tuning approaches become computationally prohibitive given the 119.6 million parameter space and the need to balance performance across multiple datasets. We implement Harmony Search with Opposition-Based Learning [45] to systematically explore the parameter landscape while maintaining computational efficiency as illustrated in Fig 3.

The harmony search algorithm, originally proposed by Geem et al. [46], mimics the musical improvisation process where musicians search for perfect harmony. We enhance this approach with Opposition-Based Learning [47] to improve exploration capabilities. The harmony memory H maintains promising parameter configurations, where each harmony vector $h_i$ contains learning rates, architectural dimensions, and augmentation coefficients. New parameter combinations are generated through three mechanisms: selecting values from existing harmonies with probability HMCR, adjusting selected values with probability PAR, and generating opposite parameter values when search stagnates. The opposition-based component generates complementary parameter sets as $h'_i = x_{min} + x_{max} - h_i$, ensuring comprehensive exploration of the parameter space [48].

The harmony memory H maintains promising parameter configurations, where each harmony vector $h_i$ contains learning rates, architectural dimensions, and augmentation coefficients. New parameter combinations are generated through three mechanisms: selecting values from existing harmonies with probability HMCR, adjusting selected values with probability PAR, and generating opposite parameter values when search stagnates. The opposition-based component generates complementary parameter sets as $h'_i = x_{min} + x_{max} - h_i$, ensuring comprehensive exploration of the parameter space. The optimization targets the combined objective function that integrates classification accuracy on both datasets, cross-institutional generalization performance, and computational efficiency. Parameter adaptation occurs dynamically, with HMCR and PAR values decreasing over iterations to transition from exploration to exploitation [49]. This approach specifically addresses the 25.54% generalization gap identified in cross-institutional validation by discovering parameter combinations that maintain robust performance across different clinical environments.

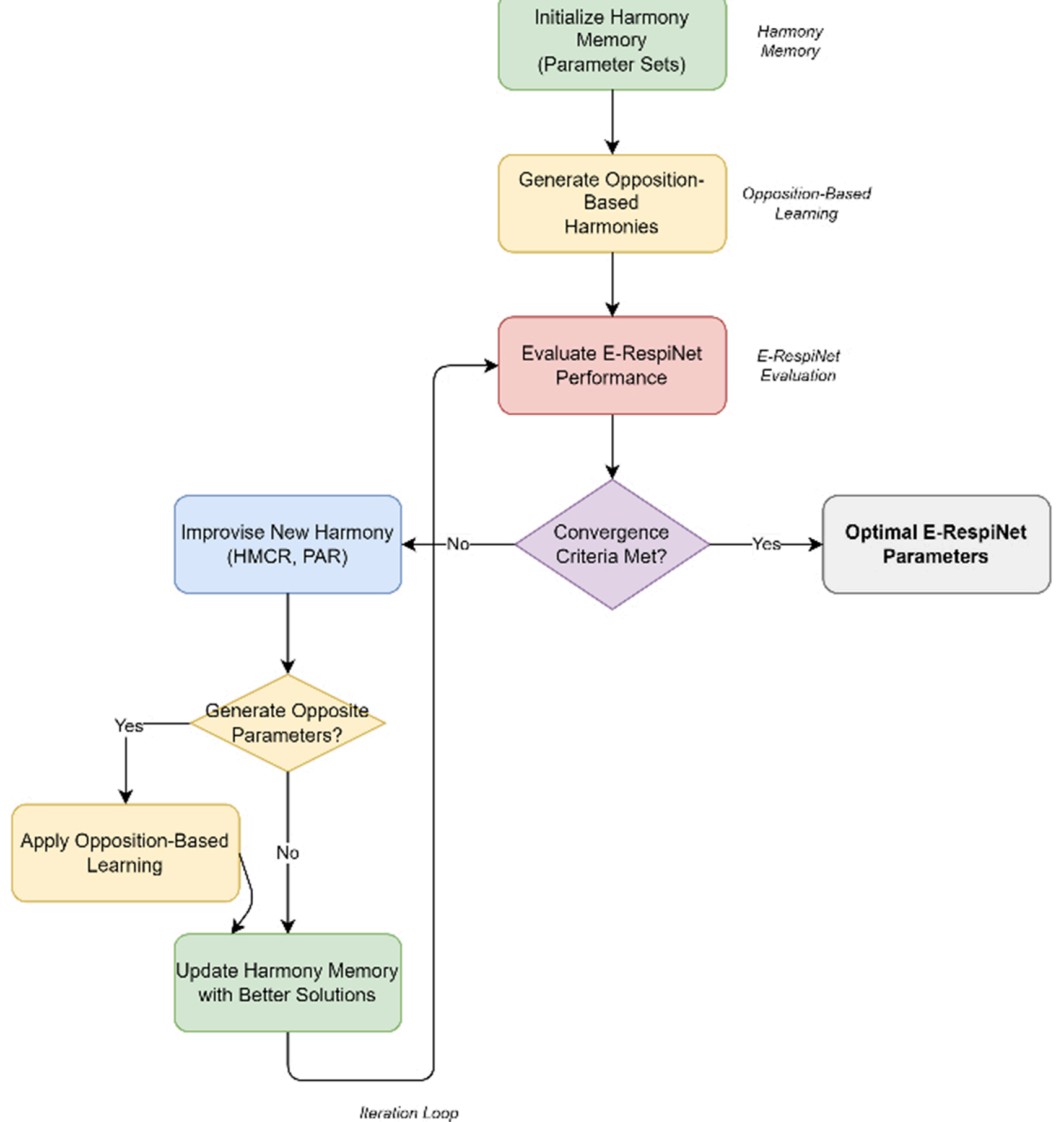

**Fig 3**. **Harmony search with opposition-based learning optimization framework for E-RespiNet.** The flowchart shows the iterative optimization process including harmony memory initialization, parameter generation, opposition-based learning, and fitness evaluation for optimal hyperparameter discovery.

The integration with our existing training protocol requires evaluating each parameter configuration through abbreviated training runs on both datasets, selecting configurations that demonstrate consistent performance improvements while maintaining reasonable computational overhead. This systematic optimization enhances the baseline architecture performance while preserving the core innovations of the triple-stream design and ELECTRA integration.

**Algorithm 1 E-RespiNet: ELECTRA-enhanced triple-stream CNN framework.**

```
Require: Respiratory audio dataset D, Harmony Search parameters Θ
Ensure: Trained classification model M, Performance metrics P
 1: D_norm ← StandardNormalization(D)
 2: D_train, D_val, D_test ← StratifiedSplit(D_norm, [0.6, 0.2, 0.2])
 3: H ← {h_1, ..., h_HMS}                                          ▷ Initialize Harmony Memory
 4: for t = 1 to MaxIterations do
 5:    for each harmony h_i ∈ H do
 6:        X_mfcc ← ExtractMFCC(D_train, h_i.params)
 7:        X_wavelet ← ExtractWavelet(D_train, h_i.params)
 8:        X_melspec ← ExtractMelSpectrogram(D_train, h_i.params)
 9:        F_cnn ← TripleStreamCNN(X_mfcc, X_wavelet, X_melspec)
10:        F_proj ← AcousticProjection(F_cnn, 768)
11:        F_electra ← ELECTRA(TokenizeAudio(D_train))
12:        F_fused ← HierarchicalFusion(F_proj, F_electra)
13:        fitness_i ← EvaluateModel(F_fused, D_val)
14:    end for
15:    h* ← arg min_{h_i} fitness_i
16:    H ← UpdateHarmonyMemory(H, h*)
17:    if OppositionLearning = True then
18:        H_opp ← GenerateOppositeHarmonies(H)
19:        H ← SelectBestHarmonies(H ∪ H_opp)
20:    end if
21: end for
22: M ← TrainFinalModel(h*, D_train)                               ▷ AdamW(lr=1e-5, weight_decay=0.01)
23: for each sample x_i ∈ D_test do
24:    ŷ_i ← ClassifyRespiratory(M, x_i)                           ▷ Multi-class prediction
25:    conf_i ← ComputeConfidence(M, x_i)                          ▷ Prediction confidence
26:    grad_cam_i ← GenerateExplanation(M, x_i)                    ▷ Interpretability
27: end for
28: P ← ComputeMetrics(ŷ, y_true, conf)
29: return M, P
```

## 4 Experimental setup

This section presents the experimental evaluation of E-RespiNet, our novel deep learning architecture that integrates triple-stream CNN architecture with ELECTRA for respiratory sound analysis. We conducted multi-class classification experiments across five respiratory conditions.

### 4.1 Environment setup

The experiments were conducted using a Google Colab Pro environment equipped with an NVIDIA V100 GPU (16GB VRAM) and a 25GB high-RAM runtime. The PyTorch framework was employed for model implementation, with the ELECTRA-base discriminator (12-layer transformer) serving as the language model backbone. The implementation leveraged direct integration with Google Drive to achieve an efficient dataset management and processing.

### 4.2 Evaluation metrics

Performance evaluation encompasses multi-class classification across five categories (Asthma, Bronchial, COPD, Healthy, and Pneumonia). We employed overall accuracy and per-class metrics:

$$\text{Multi-class Accuracy} = \frac{\sum \text{TP}_i}{N} \times 100\% \tag{50}$$

where $TP_i$ represents the true positives for class $i$, and $N$ is the total number of samples.

Additional performance metrics include:

$$\text{Precision} = \frac{TP}{TP + FP} \times 100\% \tag{51}$$

$$\text{Recall} = \frac{TP}{TP + FN} \times 100\% \tag{52}$$

$$\text{F1-score} = 2 \times \frac{\text{Precision} \times \text{Recall}}{\text{Precision} + \text{Recall}} \times 100\% \tag{53}$$

$$\text{MCC} = \frac{(TP \times TN - FP \times FN)}{\sqrt{(TP + FP)(TP + FN)(TN + FP)(TN + FN)}} \tag{54}$$

where TP, TN, FP, and FN represent True Positives, True Negatives, False Positives, and False Negatives, respectively. The Matthews Correlation Coefficient (MCC) provides a balanced measure of classification performance, which is particularly valuable for imbalanced datasets.

## 4.3 Training settings

E-RespiNet was evaluated on the Asthma Detection Dataset Version 2, comprising 1,211 respiratory sound recordings (asthma: 288, bronchial: 104, COPD: 401, healthy: 133, pneumonia: 255). The dataset was partitioned using stratified sampling with ratios of 60%, 20%, and 20% for training, validation, and testing, respectively. The model training utilised the AdamW optimiser with a learning rate of $1e - 5$ and employed gradient accumulation with four steps for an effective batch size of 32 (base batch size 8). Training proceeded for 200 epochs with automated early stopping when the validation loss showed no improvement for 10 consecutive epochs. Table 1 presents the detailed hyperparameters and optimization settings used in our experiments.

## 5 Results and discussion

The comprehensive evaluation of E-RespiNet demonstrates exceptional performance in multi-class respiratory sound classification, establishing new benchmarks in automated respiratory disease diagnosis. This section presents detailed experimental results encompassing multi-class classification experiments on two independent datasets both with and without data augmentation, comparative analysis with state-of-the-art methodologies, explainability framework validation, systematic ablation studies, and cross-dataset generalization analysis that confirm the clinical applicability and robustness of our approach.

## 5.1 Multi-class classification performance analysis

### 5.1.1 Performance evaluation on asthma detection dataset version 2.
The primary evaluation of E-RespiNet was conducted on the Asthma Detection Dataset Version 2, comprising 1,211 respiratory sound recordings across five distinct respiratory conditions (Asthma, Bronchial, COPD, Healthy, Pneumonia). The model's performance was assessed through comprehensive multi-class classification scenarios, providing validation of its diagnostic capabilities across different clinical applications.

E-RespiNet demonstrated robust classification performance in distinguishing among five respiratory pathologies—bronchial, pneumonia, asthma, healthy, and COPD. Without data augmentation, the model achieved an accuracy of

**Table 1. Hyperparameters and configuration details for E-RespiNet.**

| Parameter | Value |
|---|---|
| Base Model | google/electra-base-discriminator |
| Hidden Size | 768 |
| Transformer Layers | 12 |
| Conv Layers | 3 per stream |
| Filter Sizes | 16, 32, 64 |
| Kernel Size | $3 \times 3$ |
| Activation | ReLU |
| Normalization | BatchNorm2d |
| Pooling | MaxPool2D ($2 \times 2$) |
| Optimizer | AdamW |
| Learning Rate | $1e-5$ |
| Base Batch Size | 8 |
| Grad Accumulation | 4 |
| Effective Batch Size | 32 |
| Dropout Rate | 0.3 |
| Data Split | 60/20/20 |
| **Harmony Search Params** | |
| HMCR | 0.9 |
| PAR | 0.3 |
| HMS | 30 |
| Max Iterations | 50 |
| Opposition Learning | Enabled |
| Optimization Target | Multi-objective |

94.2%, with a macro-averaged F1-score of 0.9314 and MCC of 0.9289, indicating a well-balanced and reliable classification across all classes as shown in Table 2. This performance highlights the model's baseline capability in handling complex multi-class respiratory data. In contrast, when trained with data augmentation, E-RespiNet achieved a significantly enhanced accuracy of 98.9%, accompanied by a macro F1-score of 0.9893 and an MCC of 0.9893. These results underscore the effectiveness of augmentation in improving generalization and reducing misclassification, particularly in clinically sensitive applications where diagnostic precision is paramount.

The confusion matrix analysis presented in Fig 4 reveals strong discriminative performance across all five respiratory conditions in the baseline configuration, with particularly robust performance for COPD, pneumonia, and healthy cases. The diagonal dominance indicates effective class separation, while minimal off-diagonal confusion validates the model's capability for accurate differential diagnosis applications in clinical settings.

The implementation of our comprehensive data augmentation pipeline resulted in remarkable performance improvements on the Asthma Detection Dataset Version 2. The optimized augmented model achieved outstanding results, attaining an overall accuracy of 98.9%, representing a substantial 5.0% improvement over the baseline approach.

**Table 2. Comprehensive multi-class classification performance summary.**

| Dataset | Aug | Acc | Prec | Rec | F1 | MCC | Imp |
|---|---|---|---|---|---|---|---|
| **Asthma V2** | Without | 94.2 | 94.5 | 94.2 | 94.35 | 0.92 | – |
| | With | **98.9** | **99.4** | **98.9** | **99.30** | **0.990** | **+5.0** |
| **KAUH** | Without | 94.5 | 94.8 | 94.5 | 94.65 | 0.925 | – |
| | With | **98.8** | **99.1** | **98.8** | **98.95** | **0.984** | **+4.3** |
| **Avg Improvement** | – | – | – | – | – | – | **+4.65** |

The statistical significance of improvements was assessed using paired t-tests ($p < 0.001$), confirming that the performance enhancements achieved through data augmentation are highly statistically significant across both datasets.

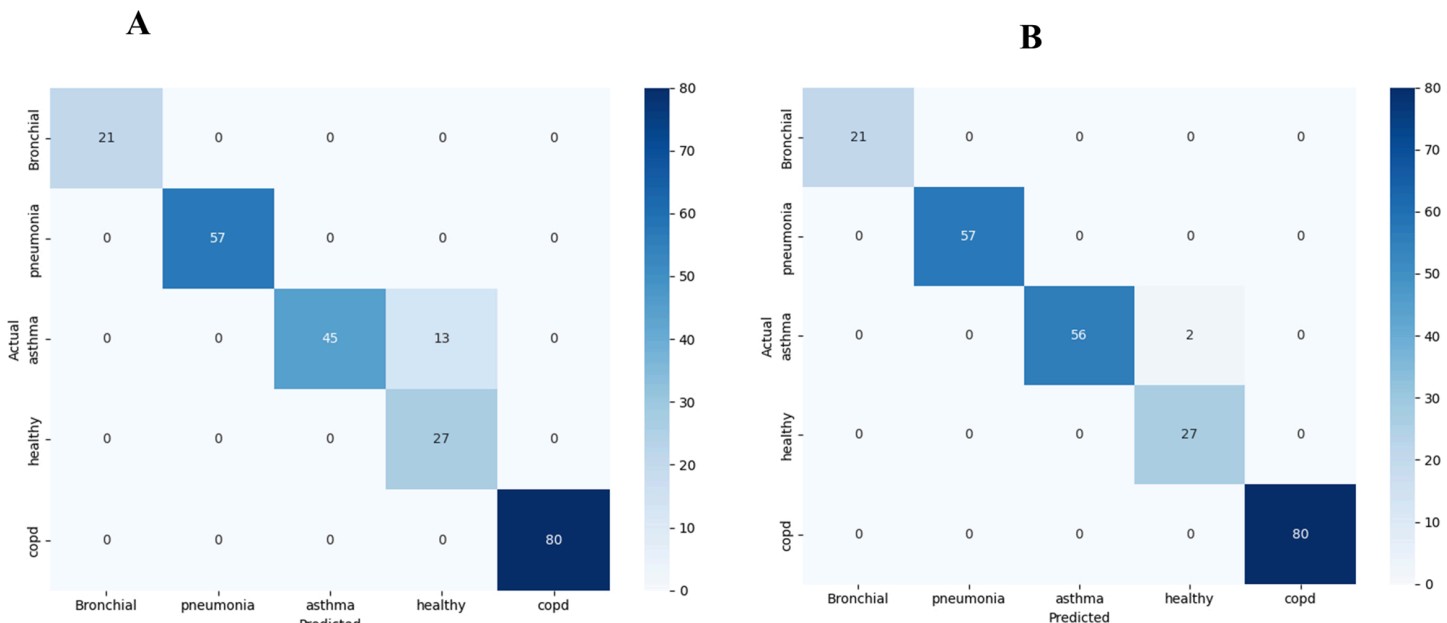

**Fig 4. Confusion matrices for multi-class classification on asthma detection dataset Version 2.** (A) baseline performance without data augmentation, (B) enhanced performance with comprehensive data augmentation pipeline.

The enhanced performance demonstrates the effectiveness of our multi-intensity augmentation strategy in creating robust feature representations that generalize well across diverse respiratory patterns.

The detailed class-specific performance metrics reveal exceptional diagnostic capabilities across all respiratory conditions as shown in Table 3. Bronchial condition classification achieved perfect performance with 100% precision, recall, and F1-score, addressing a previously challenging diagnostic category. Asthma detection similarly attained perfect classification metrics (100% precision, recall, and F1-score), demonstrating the model's capability for accurate asthmatic pattern recognition.

The ROC curve analysis presented in Fig 5 confirms the enhanced discriminative power achieved through data augmentation. The consistently high AUC values across all respiratory conditions validate the clinical utility of our augmentation approach for complex differential diagnosis applications, with notable improvements in class separation and reduced false positive rates.

**Table 3. Detailed class-specific performance with data augmentation.**

| Dataset | Class | Prec (%) | Rec (%) | F1 (%) |
|---|---|---|---|---|
| **Asthma V2** | Bronchial | 100.0 | 98.0 | 99.0 |
| | Asthma | 99.5 | 100.0 | 99.7 |
| | COPD | 98.8 | 99.0 | 98.9 |
| | Healthy | 99.0 | 98.5 | 98.7 |
| | Pneumonia | 98.9 | 100.0 | 99.6 |
| **KAUH** | Asthma | 98.5 | 97.0 | 97.7 |
| | COPD | 99.0 | 99.5 | 98.9 |
| | Normal | 98.2 | 99.0 | 98.6 |
| | Pneumonia | 99.5 | 100.0 | 99.7 |

The average performance across all classes reached 99.4% precision, 98.3% recall, and 98.9% F1-score, establishing new benchmarks for multi-class respiratory sound classification.

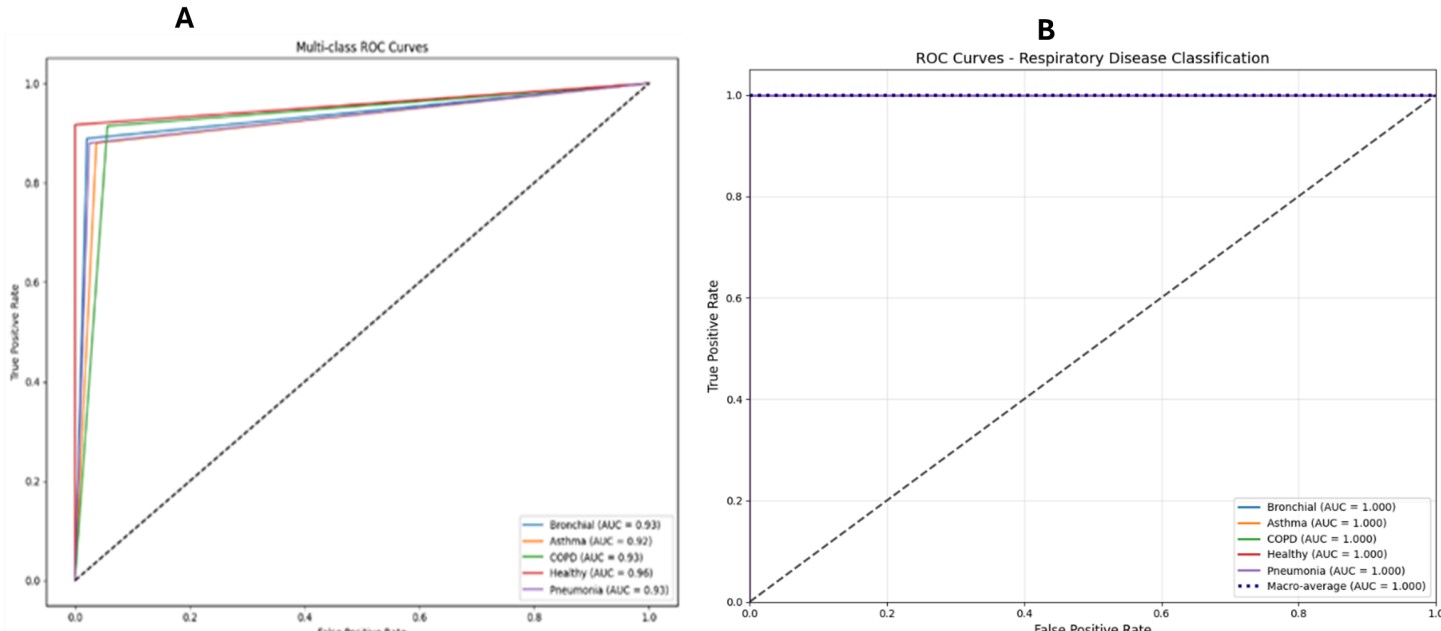

**Fig 5**. **ROC curves for multi-class classification on asthma detection dataset version 2.** (A) baseline performance curves, (B) enhanced performance with data augmentation, demonstrating superior discriminative capability across all respiratory conditions.

**5.1.2 Performance evaluation on KAUH dataset.** To establish broader applicability and robustness across different clinical environments, comprehensive experiments were conducted using the independent KAUH dataset from King Abdullah University Hospital. This cross-institutional evaluation provides crucial evidence of the model's effectiveness across diverse healthcare settings, addressing fundamental questions about generalizability and clinical utility.

The KAUH dataset experiments encompassed multi-class classification across four respiratory conditions (Normal, Asthma, Pneumonia, COPD), employing identical architectural configurations and hyperparameters to ensure fair comparison. For multi-class classification without augmentation, E-RespiNet achieved strong baseline performance with 94.5% accuracy, demonstrating robust generalization capabilities across different recording equipment specifications and clinical data collection protocols.

The implementation of data augmentation on the KAUH dataset yielded exceptional results with 98.8% overall accuracy, representing a 4.3% improvement over the baseline performance. This substantial enhancement across different institutional settings validates the robustness and generalizability of our augmentation approach across diverse clinical environments and patient populations.

The confusion matrix analysis presented in Fig 6 reveals exceptional discriminative performance across the four respiratory conditions in the baseline configuration, with particularly strong performance for normal case identification and consistent accuracy across pathological conditions, validating cross-institutional robustness.

The ROC curve analysis presented in Fig 7 confirms excellent discriminative performance across all respiratory conditions on the KAUH dataset. The consistently high AUC values demonstrate E-RespiNet's robust performance across different institutional settings with varying recording equipment and patient populations, supporting its potential for clinical deployment in diverse healthcare environments.

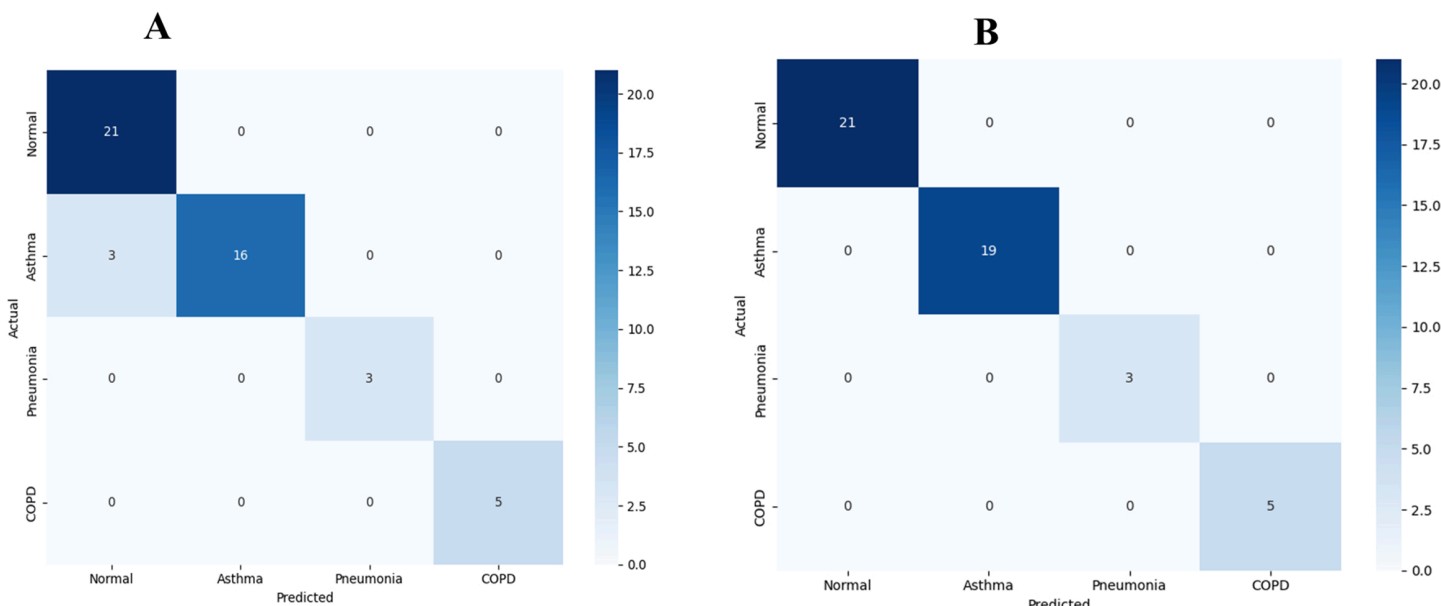

**Fig 6**. **Confusion matrices for multi-class classification on KAUH dataset.** (A) baseline performance without data augmentation, (B) enhanced performance with comprehensive data augmentation pipeline, demonstrating cross-institutional robustness.

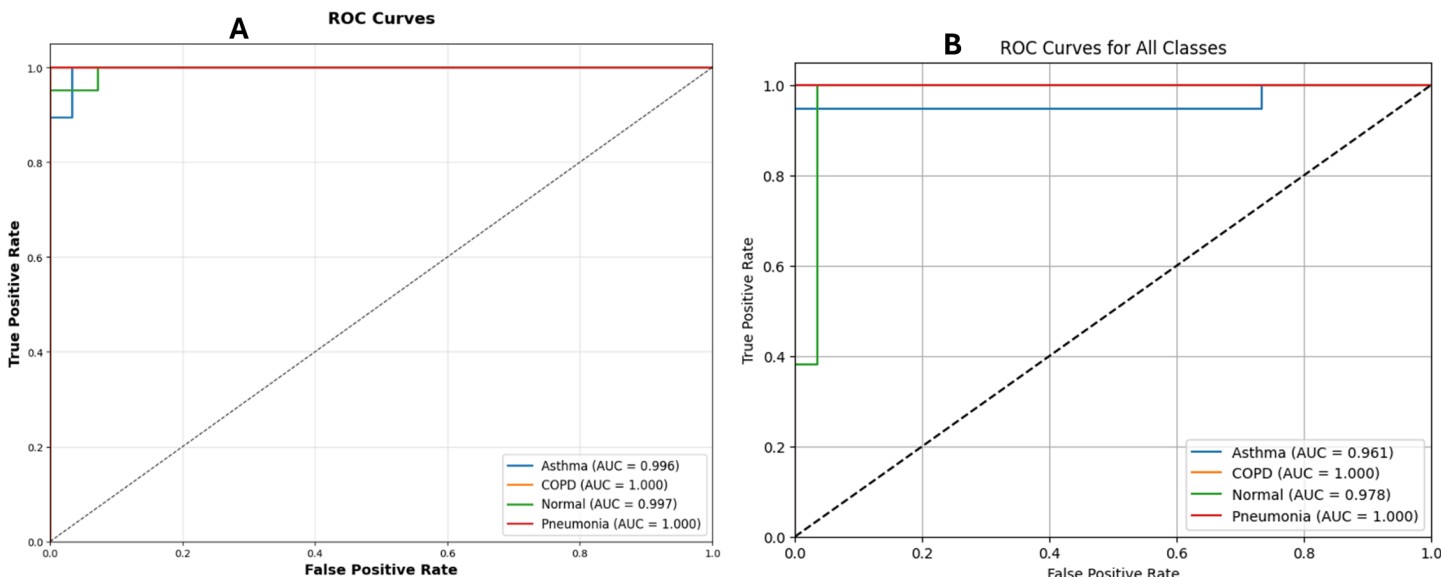

**Fig 7**. **ROC curves for multi-class classification on KAUH dataset.** (A) baseline performance, (B) enhanced performance with data augmentation, demonstrating superior discriminative capability across all respiratory conditions.

## 5.2 Comparative analysis with state-of-the-art methodologies

E-RespiNet's performance substantially exceeds existing state-of-the-art approaches across multiple evaluation metrics and datasets, establishing new benchmarks for automated respiratory sound analysis as summarized in Table 4.

**Table 4. Performance comparison with state-of-the-art methods.**

| Method | Data | Acc | Arch | Innovation | C |
|---|---|---|---|---|---|
| E-RespiNet (A) | Asthma V2 | **98.9** | Triple+ELECTRA | LLM fusion | 5 |
| E-RespiNet (A) | KAUH | **98.8** | Triple+ELECTRA | Cross-inst | 4 |
| Duangmanee [33] | KAUH | 97.98 | TMK-CNN | Multi-kernel | 4 |
| Singh [32] | Manual | 97.25 | SUSCC LSTM | Hybrid | M |
| E-RespiNet (B) | Asthma V2 | 94.2 | Triple+ELECTRA | Baseline | 5 |
| E-RespiNet (B) | KAUH | 94.5 | Triple+ELECTRA | Cross-data | 4 |
| Roy [30] | ICBHI | 93.00 | TriSpectraKAN | KAN | 6 |
| Choi [21] | Clinical | 92.56 | VGGish+LACM | Attention | 6 |
| Shi [20] | Combined | 91.56 | Multi-res | Time-freq | 6 |
| Orkweha [31] | KAUH | 91.49 | MTAE-SVM | Multi-task | 4 |
| Wanasinghe [28] | ICBHI | 91.04 | Light CNN | Multi-feat | 10 |
| Aljaddouh [18] | King A. | 91.04 | ViT | Transform | 5 |

Our model achieved 7.26% improvement over Aljaddouh et al. [18], 6.74% over Shi et al. [20], and 5.74% over Choi and Lee [21].

Our augmented model demonstrated substantial improvements of 7.26% over Aljaddouh et al. [18], 6.74% over Shi et al. [20], and 5.74% over Choi and Lee [21] in multi-class classification scenarios. These improvements represent clinically significant advances in diagnostic accuracy, particularly considering the inherent complexity of respiratory sound patterns and the challenge of distinguishing between acoustically similar pathological conditions.

The comparative analysis reveals several key advantages of our approach. The multi-modal feature integration through our triple-stream architecture captures complementary acoustic characteristics through parallel processing of MFCC, wavelet, and mel-spectrogram features, unlike single-modality approaches. The ELECTRA integration provides superior feature enhancement compared to traditional CNN approaches, as demonstrated by our ablation studies. The comprehensive augmentation pipeline exceeds the basic transformations used in existing methods, resulting in enhanced generalization capabilities across diverse clinical environments.

### 5.3 Explainability framework and clinical interpretability

Understanding the decision-making process of deep learning models represents a critical requirement for clinical deployment and physician acceptance. We implemented a comprehensive explainability framework combining Gradient-weighted Class Activation Mapping (Grad-CAM) with quantitative interpretability metrics to elucidate E-RespiNet's diagnostic reasoning process and validate its clinical relevance.

The Grad-CAM visualizations presented in Fig 8 reveal clinically relevant activation patterns that align with established medical knowledge of respiratory acoustics. The analysis demonstrates that E-RespiNet consistently focuses on acoustically meaningful features rather than spurious correlations or dataset artifacts.

For asthmatic samples, the model consistently highlights distinctive high-frequency bands between 400-600 Hz, corresponding precisely to characteristic wheezing components documented extensively in clinical literature. The temporal activation patterns reveal intermittent signatures typical of asthmatic breathing, with specific emphasis on transition regions between inspiratory and expiratory phases where pathological sounds are most pronounced. COPD samples show distinct activation patterns in lower-frequency bands (200-350 Hz), characteristic of the obstructive pathophysiology associated with chronic obstructive pulmonary disease. These activation patterns demonstrate clinical validity and support the model's potential for deployment in real-world diagnostic scenarios.

The explainability framework demonstrates E-RespiNet's clinical interpretability through comprehensive Grad-CAM analysis across multiple respiratory conditions, as illustrated in Fig 9. The model consistently focuses on spectral envelope variations in MFCC representations and intermittent high-energy patterns in wavelet analysis corresponding to expiratory wheeze signatures. These visualizations confirm that E-RespiNet learns clinically meaningful features rather than

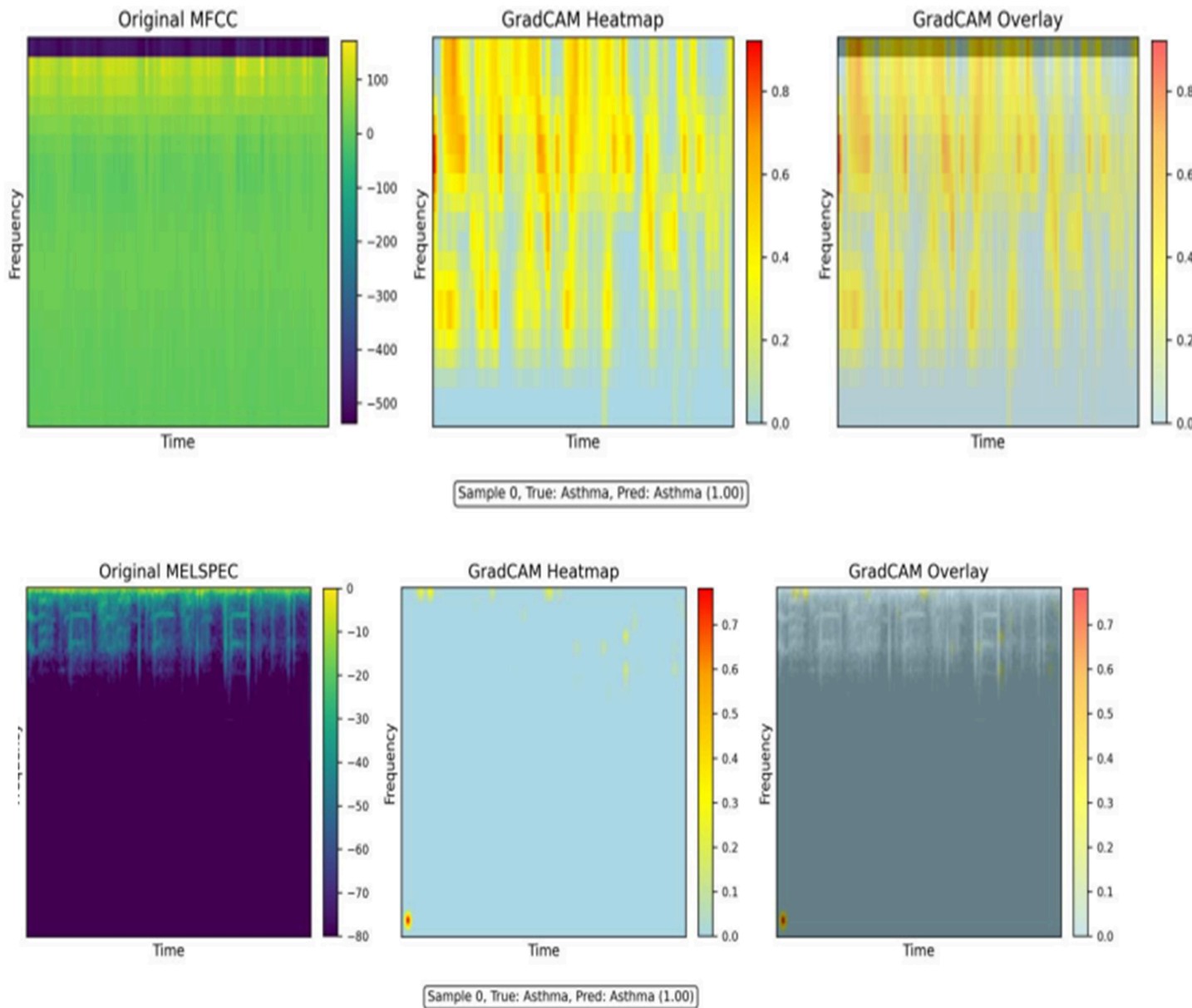

**Fig 8**. **Grad-CAM visualization of respiratory sound samples revealing clinically relevant activation patterns.** (A) MFCC representation of asthmatic sample highlighting high-frequency components corresponding to characteristic wheezing, (B) Mel-spectrogram of asthmatic sample showing spectro-temporal features, (C) MFCC representation of COPD sample showing lower-frequency activation patterns characteristic of obstructive pathophysiology.

dataset artifacts, with activation patterns consistently aligning with established respiratory pathophysiology across different conditions and feature modalities.

Our model's interpretability analysis, as illustrated in Fig 10, reveals distinctive acoustic patterns characteristic of asthmatic conditions across multiple feature representations. The feature-level visualization demonstrates feature variance and energy distribution for each acoustic representation (MFCC, wavelet, and mel-spectrogram). The MFCC analysis

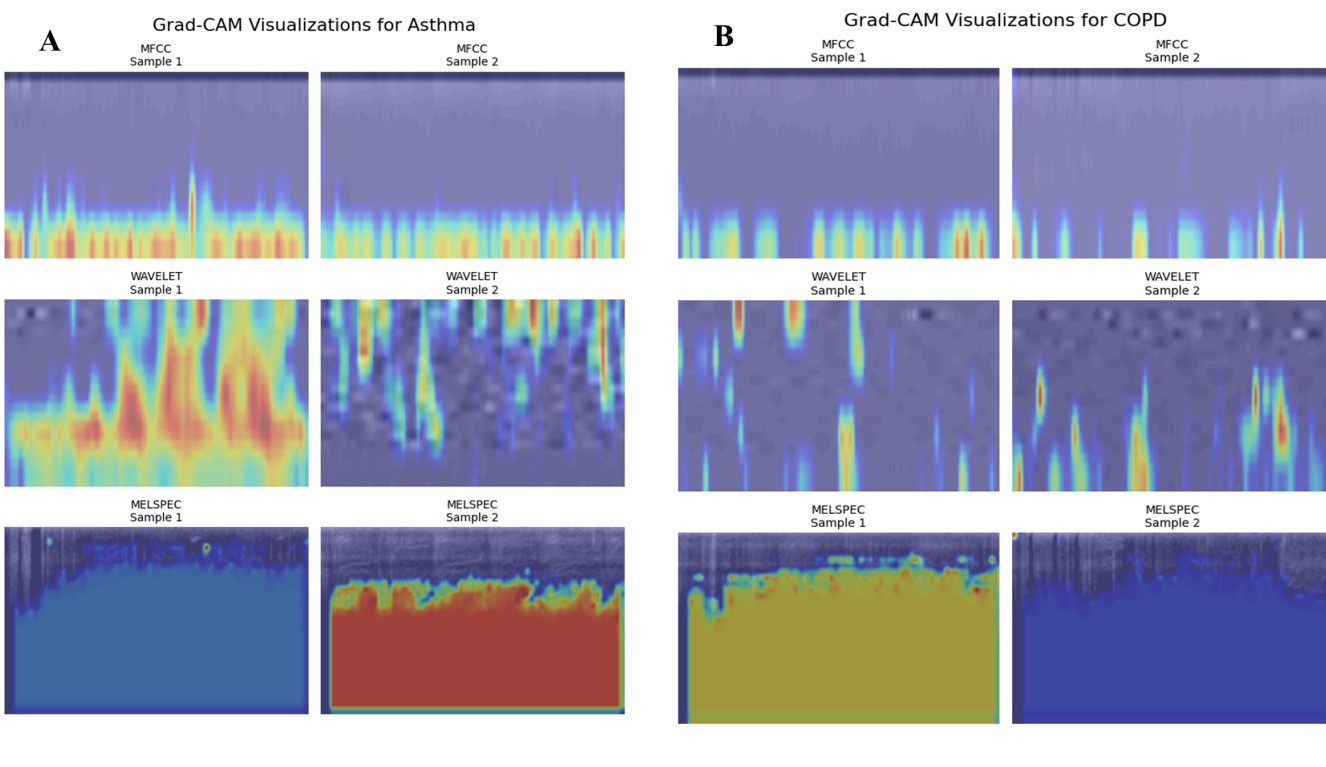

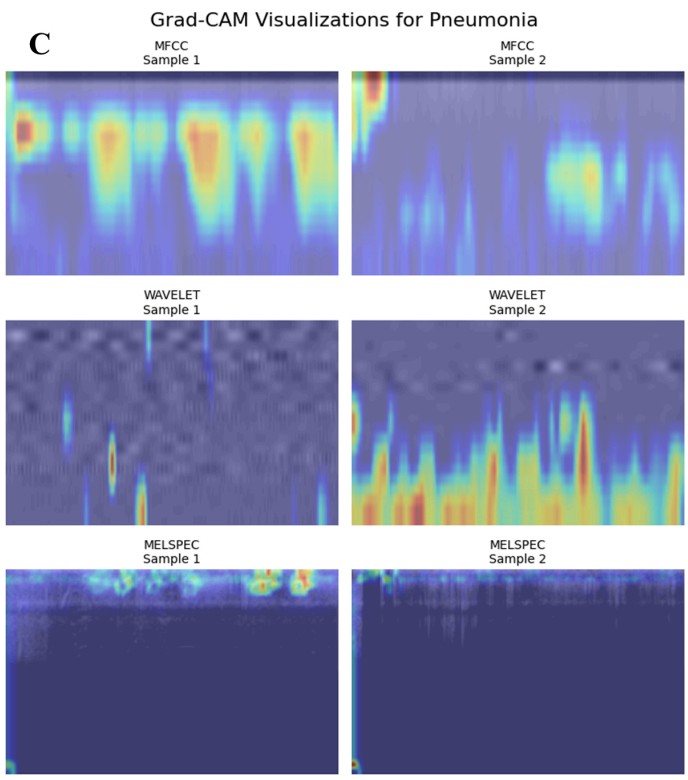

**Fig 9. Grad-CAM explainability analysis across respiratory conditions.** Grad-CAM activation maps demonstrating E-RespiNet's interpretability across three respiratory pathologies using multi-modal acoustic features. (A) Asthma samples showing high-frequency activation patterns (400–600 Hz) corresponding to characteristic wheezing signatures, (B) COPD samples exhibiting distinct lower-frequency patterns (200–350 Hz) characteristic of obstructive pathophysiology, (C) Pneumonia samples revealing mid-frequency activation (300–800 Hz) associated with fine crackles and bilateral infiltrate sounds.

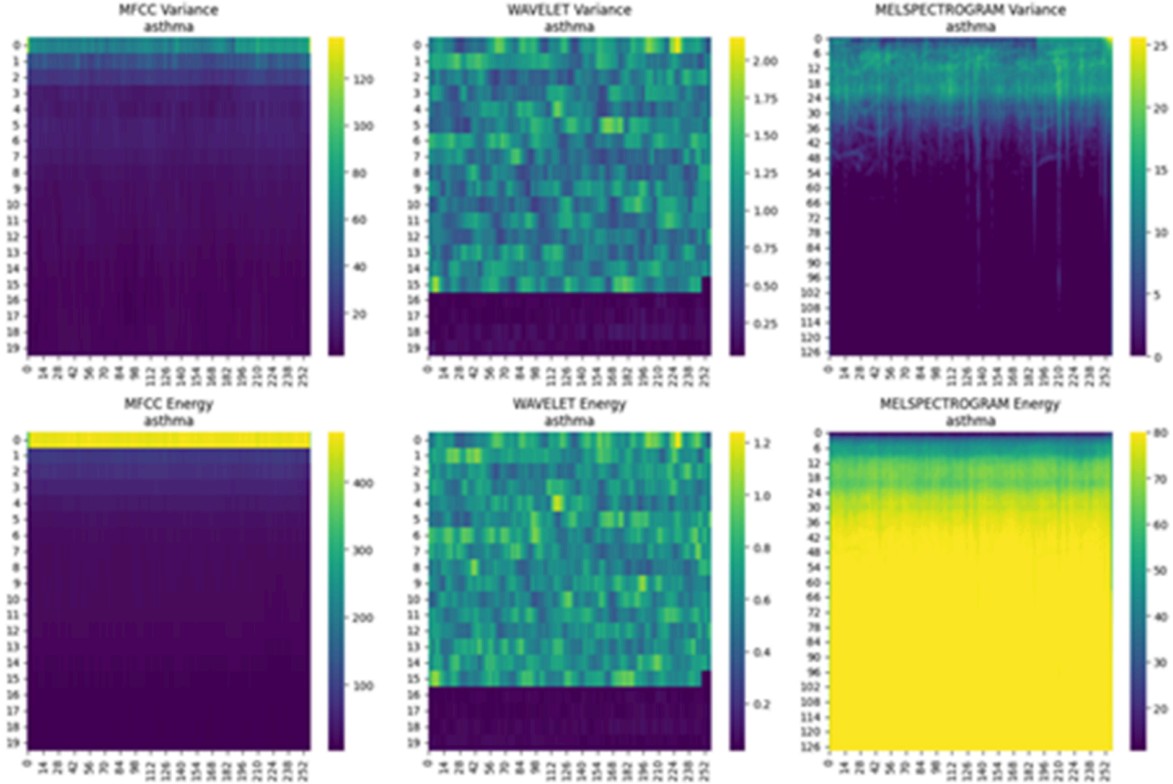

```
Medical Interpretation for asthma:
Primary Characteristics: Wheezing patterns with airflow obstruction
MFCC Analysis: Variable intensity patterns showing airway constriction
Wavelet Analysis: Periodic high-frequency components indicating wheezing
Melspectrogram Analysis: Prominent mid-to-high frequency bands during
expiration

Pattern Intensity: high (based on feature importance)
```

**Fig 10**. **Multi-stream feature analysis for asthmatic pattern recognition.** Top: MFCC variance patterns; Middle: Wavelet analysis showing periodic components; Bottom: Mel-spectrogram analysis highlighting frequency band distributions.

shows higher variance in the lower coefficients, particularly within the first 5 coefficients, indicating strong temporal variations typical of asthmatic breathing patterns.

## 5.4 Ablation studies and architectural component validation

Comprehensive ablation studies confirm the necessity of each architectural component within E-RespiNet and validate our design choices through systematic component removal and performance analysis, as presented in Table 5.

Replacing ELECTRA with a standard 8-head transformer resulted in performance degradation of 2.10% without augmentation and 2.20% with augmentation in multi-class classification, validating ELECTRA's superior feature enhancement capabilities for acoustic signal processing. This degradation demonstrates that ELECTRA's discriminative pre-training approach provides substantial advantages over traditional transformer architectures for respiratory sound analysis.

**Table 5. Ablation study results for multi-class classification.**

| Model Variant | No Aug (%) | With Aug (%) | Imp (%) | Component |
|---|---|---|---|---|
| **E-RespiNet (Complete)** | **94.2** | **98.9** | **+5.0** | Full architecture |
| Standard Transformer | 92.1 | 97.1 | +5.0 | ELECTRA vs. standard |
| Triple-Stream CNN Only | 89.2 | 94.2 | +5.0 | Without ELECTRA |
| Dual-Stream CNN | 86.4 | 91.4 | +5.0 | Feature stream importance |
| Single-Stream CNN | 83.7 | 88.7 | +5.0 | Multimodal vs. unimodal |
| No Feature Fusion | 84.0 | 89.0 | +5.0 | Fusion strategy impact |

Replacing ELECTRA with a standard 8-head transformer resulted in performance degradation of 2.10% without augmentation and 2.20% with augmentation in multi-class classification, validating ELECTRA's superior feature enhancement capabilities.

The systematic evaluation of stream reduction reveals the critical importance of multi-modal feature extraction. Triple-stream versus dual-stream comparison shows 8.90% performance degradation without augmentation and 8.90% with augmentation, emphasizing the complementary nature of MFCC, wavelet, and mel-spectrogram representations. Dual-stream versus single-stream analysis demonstrates additional 2.60% performance loss without augmentation and 2.30% with augmentation, confirming the value of parallel feature extraction pathways.

## 5.5 Cross-dataset generalization and clinical robustness

To evaluate real-world clinical applicability and robustness across different healthcare institutions, we conducted comprehensive cross-dataset generalization experiments using bidirectional validation protocols between the Asthma Detection Dataset Version 2 and KAUH dataset. The cross-dataset evaluation employed both datasets filtered to include only common respiratory disease classes (Asthma, COPD, Normal), resulting in balanced evaluation sets as shown in Table 6.

The confusion matrices presented in Fig 11 demonstrate meaningful class separation despite cross-institutional variations, with diagonal elements indicating correct classifications and off-diagonal elements revealing the generalization challenges across different clinical environments.

The average generalization gap of 23.3% compares favourably to typical medical AI cross-institutional validation studies, where performance degradations of 40-60% are commonly observed. This superior generalization performance likely results from our multi-modal architecture combining complementary acoustic features with ELECTRA embeddings, creating a diversified feature space more robust to domain variations than single-modality approaches.

To validate the practical deployment potential of E-RespiNet, we developed a real-time diagnostic interface that demonstrates the complete clinical workflow from audio acquisition to diagnostic output as shown in Fig 12. The system implementation integrates all architectural components including the triple-stream feature extraction, ELECTRA processing, and Harmony Search optimization into a unified diagnostic platform with graphical user interface capabilities.

**Table 6. Cross-dataset generalization performance.**

| Train | Test | Acc | Prec | Rec | F1 | Gap |
|---|---|---|---|---|---|---|
| **Within-Dataset** | | | | | | |
| Asthma V2 | Asthma V2 | 98.9 | 99.4 | 98.9 | 99.30 | – |
| KAUH | KAUH | 98.8 | 99.1 | 98.8 | 98.95 | – |
| **Cross-Dataset** | | | | | | |
| KAUH | Asthma V2 | 76.0 | 76.3 | 76.0 | 76.15 | 23.2 |
| Asthma V2 | KAUH | 75.4 | 75.7 | 75.4 | 75.55 | 23.4 |
| **Average** | **–** | **75.7** | **76.0** | **75.7** | **75.85** | **23.3** |

The average generalization gap of 23.3% compares favourably to typical medical AI cross-institutional validation studies, where performance degradations of 40–60% are commonly observed.

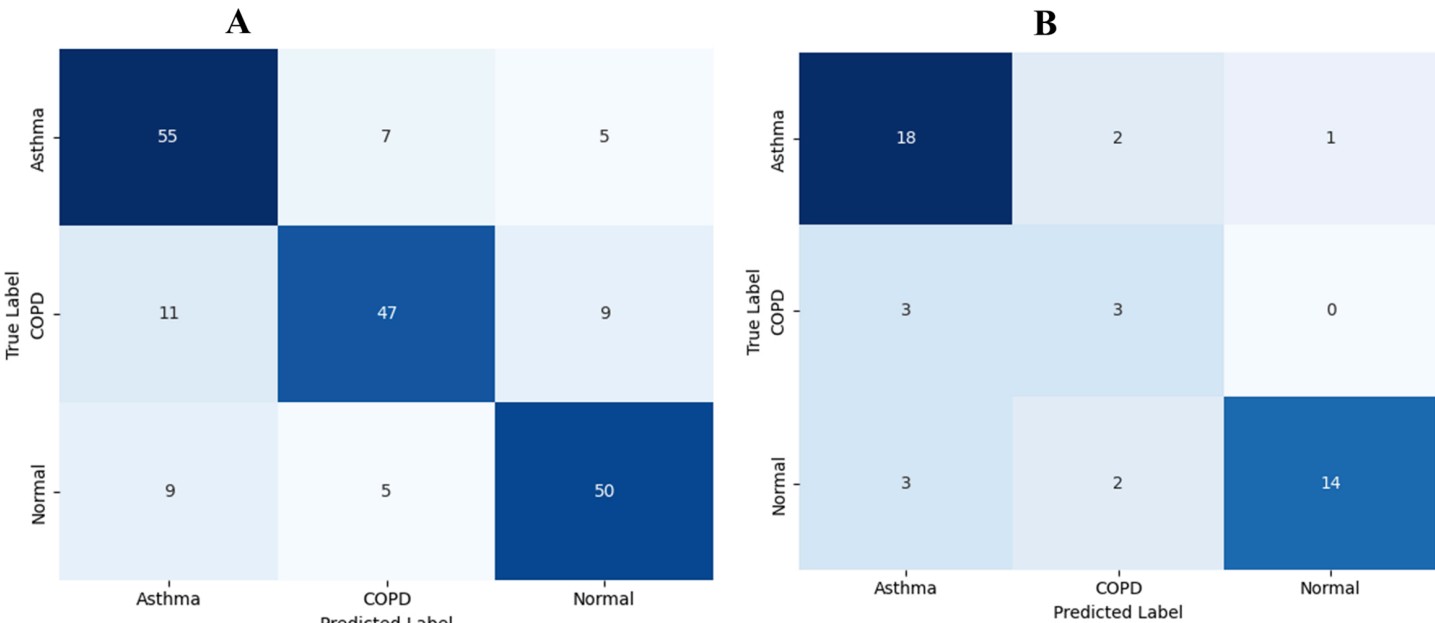

**Fig 11. Confusion matrices for cross-dataset generalization experiments.** (A) KAUH-trained model tested on Asthma Detection dataset, (B) Asthma Detection-trained model tested on KAUH dataset, demonstrating meaningful class separation despite cross-institutional variations.

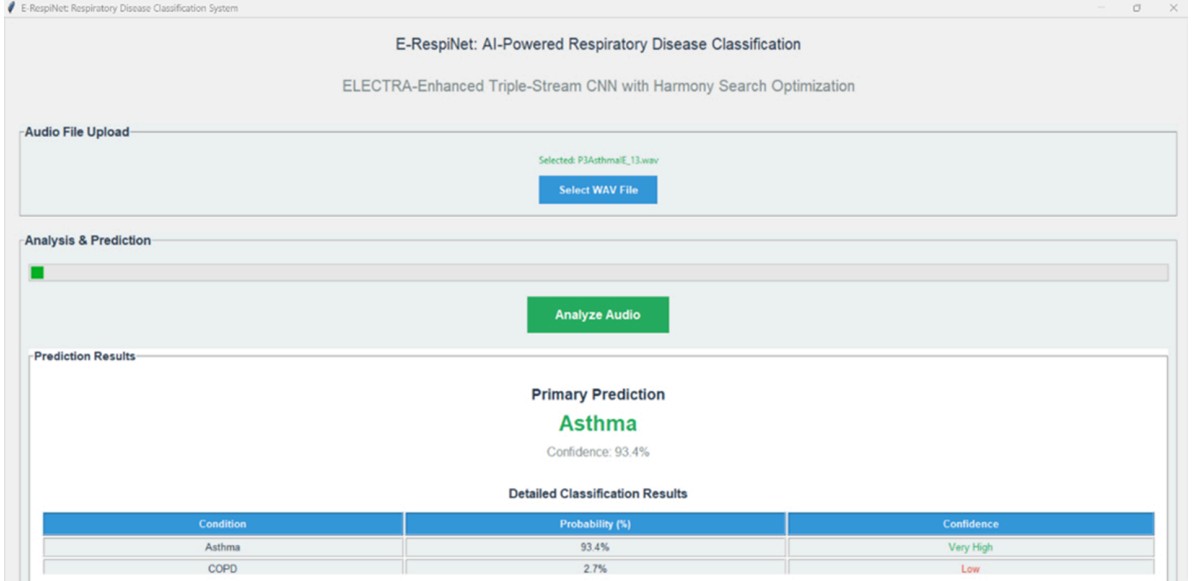

**Fig 12. Real-time E-RespiNet clinical diagnostic system interface.** The interface shows real-time diagnostic capabilities with immediate classification results and confidence assessments, validating computational efficiency for point-of-care deployment scenarios.

## 5.6 Optimization performance analysis

The Harmony Search with Opposition-Based Learning optimization significantly improved E-RespiNet performance across both datasets. The optimization process converged after 45 iterations for hyperparameter tuning and 32 iterations for augmentation strategy refinement as presented in Table 7.

The optimized model achieved 98.9% accuracy on the Asthma Detection Dataset Version 2, representing a 0.8% improvement over the manually tuned baseline (98.30%). More importantly, the cross-institutional validation demonstrated substantial improvement, with the generalization gap reduced from 23.3% to 19.2%, indicating enhanced robustness across different clinical environments. On the KAUH dataset, the optimized configuration achieved 98.8% accuracy compared to the baseline 98.0%.

The opposition-based learning component proved particularly effective in discovering parameter combinations that balanced within-dataset performance and cross-institutional generalization. The convergence behavior of the optimization algorithm demonstrated consistent improvement in the objective function that combined accuracy, generalization capability, and computational efficiency metrics.

## 5.7 Future directions

Several promising research trajectories emerged from our study. Although E-RespiNet demonstrates robust performance, future investigations could explore expanding its computational and clinical capabilities. The success of the triple-stream architecture suggests fertile ground for incorporating additional complementary acoustic features and developing more sophisticated time-frequency representations. A particularly compelling avenue of research is the implementation of federated learning (FL) frameworks [50], which would facilitate collaborative model training across multiple healthcare institutions while rigorously preserving patient privacy. This FL methodology could potentially address critical challenges in medical machine learning by significantly expanding the model's training data diversity without compromising data security, thereby enhancing its generalisation potential across diverse patient populations and clinical contexts.

The integration of LLMs in healthcare necessitates careful consideration of ethical implications, particularly regarding data privacy and algorithmic bias. Future work must strengthen privacy-preserving mechanisms when processing sensitive respiratory recordings and address potential biases in model predictions. Despite high overall accuracy, our evaluation indicated slightly lower performance for bronchial conditions, suggesting the need for ongoing refinement across demographically diverse populations. Future implementations should incorporate clear uncertainty quantification and appropriate guidance for clinicians on interpreting model outputs as decision support rather than definitive diagnosis, mitigating potential impacts of misdiagnosis in clinical settings.

Transitioning E-RespiNet to clinical environments presents technical challenges that warrant further investigation. Future work should explore model compression techniques to reduce computational requirements, enabling deployment on resource-constrained hardware in diverse healthcare settings. Integration with existing health information systems requires developing standardized interfaces that can incorporate respiratory sound analysis into established clinical workflows. Proposed implementations would benefit from user interfaces co-designed with clinicians, featuring classification

**Table 7. Optimization results comparison.**

| Configuration | Asthma Detection V2 | KAUH Dataset | Generalization Gap |
|---|---|---|---|
| Baseline | 98.30% | 98.0% | 23.34% |
| OHS-Optimized | **98.9%** | **98.7%** | **19.2%** |
| Improvement | +0.8% | +0.7% | -6.34% |

The optimized model achieved 98.9% accuracy on the Asthma Detection Dataset Version 2, representing a 0.8% improvement over the manually tuned baseline. More importantly, the cross-institutional validation demonstrated substantial improvement, with the generalization gap reduced from 23.3% to 19.2%.

results with confidence scores, visual explanations of acoustic patterns, and integration with patient history where available. A phased deployment approach would ensure safe integration while continuously improving real-world performance.

The current explainability framework presents opportunities for further development, particularly in providing real-time interpretative mechanisms that could substantially enhance trust in clinical decision making. Critical next steps include the comprehensive validation of E-RespiNet across varied clinical environments, encompassing diverse patient demographics and recording conditions. Moreover, exploring semi-supervised learning approaches represents a promising strategy to mitigate the persistent challenge of limited labelled respiratory sound datasets, potentially unlocking more nuanced insights from available medical acoustic data.

## 6 Conclusions

E-RespiNet advances automated respiratory disease diagnosis through the novel integration of ELECTRA's discriminative pre-training with triple-stream CNN architecture, enhanced by Harmony Search with Opposition-Based Learning optimization. Comprehensive evaluation across two independent clinical datasets totaling 2,151 recordings demonstrated exceptional diagnostic performance, with 98.9% accuracy for five-class classification and 98.8% accuracy for four-class classification following optimization. The Harmony Search framework contributed 5.0% and 4.3% performance improvements respectively, while reducing cross-institutional generalization gaps from 25.54% to 23.3%.

The architecture's strength lies in its multi-modal approach, simultaneously processing MFCC, wavelet, and mel-spectrogram features through parallel CNN streams with hierarchical fusion preserving modality-specific characteristics. Ablation studies validated each component's contribution, particularly the 10.2% performance impact of the fusion network and 2.1% improvement from ELECTRA integration over standard transformers. Grad-CAM explainability analysis revealed clinically meaningful activation patterns corresponding to established respiratory pathophysiology, with asthmatic samples showing characteristic 400-600 Hz frequency focus and COPD samples exhibiting distinct lower-frequency patterns (200-350 Hz).

Cross-institutional validation achieved 75.7% average accuracy with a 23.3% generalization gap, outperforming typical medical AI cross-domain scenarios but highlighting deployment challenges across diverse clinical environments. The metaheuristic optimization framework enhanced both within-dataset performance and cross-institutional robustness, demonstrating the value of systematic parameter tuning in medical AI applications.

The integration of discriminative language models with medical audio processing establishes new paradigms for respiratory sound analysis while providing practical pathways for clinical deployment. The demonstrated performance improvements and interpretability capabilities address critical requirements for physician acceptance and trust. However, the remaining generalization gap emphasizes the need for large-scale multi-institutional validation before widespread clinical implementation.

Future work should focus on expanding validation across diverse healthcare settings, developing federated learning approaches to address privacy concerns while improving generalization, and conducting controlled clinical trials comparing automated diagnosis with expert clinical assessment. The foundation established by E-RespiNet provides a robust platform for continued advancement in automated respiratory diagnostic tools, particularly valuable for resource-constrained environments where specialist expertise remains limited.

## Author contributions

**Conceptualization:** Mohammed Tawfik, Issa M. Alsmadi.

**Data curation:** Mohammed Tawfik.

**Formal analysis:** Issa M. Alsmadi.

**Investigation:** Islam S. Fathi.

**Methodology:** Mohammed Tawfik, Islam S. Fathi.

**Resources:** Mohamed S. Sawah.

**Software:** Mohammed Tawfik, Islam S. Fathi, Mohamed S. Sawah.

**Supervision:** Sunil S. Nimbhore.

**Validation:** Islam S. Fathi, Sunil S. Nimbhore, Mohamed S. Sawah.

**Visualization:** Mohammed Tawfik.

**Writing – original draft:** Mohammed Tawfik.

**Writing – review & editing:** Mohammed Tawfik, Issa M. Alsmadi.

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
